# Higher momentary parental burnout predicts lower subsequent emotional expression in parents during the festive season
Ziwen Teuber [1] ✉, Elouise Botes[1,2], Julia Reiter[3], Samuel Greiff[4,5], Kaisa Aunola [6] & Daniel McNeish[7]

This study adopted a within-person lens to unpack parental burnout and genuine emotional expression, focusing on their interplay and dynamic patterns – inertia, variability, and person-specific mean – during the Christmas season, an emotionally charged period that offers a valuable time window to study affective dynamics in parenting. Using the experience sampling method, we conducted a 35-day real-time study with 293 U.K. parents (14,451 observations), supplemented by baseline and follow-up assessments. Dynamic structural equation modeling was used to test reciprocal within-person relations between both constructs over time, to assess individual differences in dynamic patterns, and to explore whether these patterns mediated changes in burnout and expression from baseline to follow-up. Results revealed a negative, unidirectional within-person association from parental burnout to genuine expression. Individual differences were found in inertia, variability, and person-specific mean levels for both constructs. Notably, these person-specific mean levels mediated the links between baseline and follow-up levels of parental burnout and genuine expression. These findings offer insights into how short-term dynamics in parental burnout and genuine expression shape longer-term affective (mal)adjustment. They suggest that future intervention programs could benefit from being personalized and delivered in real time, targeting emotion regulation and burnout recovery in parents, particularly during emotionally intense periods such as the holiday season.

At the intersection of parenting and affective science, the link between parental burnout and emotion regulation has garnered growing scholarly attention, driven by the alarming prevalence of parental burnout[1,2] and the profound influences of emotion regulation on mental health[3–5]. However, the directionality of effects between these variables remains empirically inconsistent, particularly when differentiating between within- and between-person levels[4,6–8]. Critically, both parental burnout and emotion regulation are inherently dynamic and unfold over time. Scholars in affective science have identified inertia (resistance to change) and variability (the magnitude of fluctuations) as two distinct within-person dynamic patterns that offer complementary insights into psychological functioning and

mental health[9–13]. Despite this growing recognition, two critical gaps remain. First, we still know little about the moment-to-moment, within-person associations between parental burnout and emotion regulation, i.e., how these two constructs co-vary in real time. Second, although recent work has begun to explore within-person dynamics in parental burnout[6], no study to date has simultaneously examined inertia and variability in both parental burnout and emotion regulation. Addressing these research gaps is essential to better understand the micro-level affective processes that may underlie longer-term patterns of adjustment or risk in parenting. This study aimed to address these gaps by focusing on the dynamic interplay between parental burnout and genuine emotional expression, an often-overlooked construct

[1]Department of Behavioral and Cognitive Sciences, University of Luxembourg, Esch-sur-Alzette, Luxembourg. [2]Luxembourg Centre for Educational Testing, University of Luxembourg, Esch-sur-Alzette, Luxembourg. [3]Department of Occupational, Economic, and Social Psychology, University of Vienna, Vienna, Austria. [4]School of Social Sciences and Technology, Technical University of Munich, Munich, Germany. [5]Centre for International Student Assessment (ZIB), Technical University of Munich, Munich, Germany. [6]Department of Psychology, University of Jyväskylä, Jyväskylä, Finland. [7]Department of Psychology, Arizona State University, Tempe, USA. ✉e-mail: ziwen.teuber@uni.lu

in parenting literature. Parental genuine expression is defined as the open yet appropriate communication of one's true feelings to their children, regardless of affective valence[14,15]. Drawing on emotional labor[14,16,17] and emotion socialization[18] frameworks, we suggest that, when enacted deliberately and skillfully, genuine expression serves as an adaptive form of emotion regulation in parent-child interactions, fulfilling both parent- and child-focused functions. We applied a dynamic structural equation modeling (DSEM)[19,20] approach to 35-day intensive longitudinal real-time data of 293 UK parents, complemented by baseline and follow-up measures across the Christmas season. While Christmas promotes family togetherness and cherished moments, it is also accompanied by social and logistical demands that can place additional stress on parents[21], providing it a valuable window into the dynamics of parents' affective experiences.

Parental burnout is a multidimensional construct, characterized by emotional exhaustion, where parents feel drained in their parental role; emotional distancing, marked by a sense of detachment or disconnection from their children; a sense of being fed up, where parents struggle to meet anything beyond their children's basic needs; and a contrast to their previous parental self, manifesting in feelings of inadequacy compared to how they once perceived themselves as parents[22]. Parental burnout has been associated with more severe mental issues, including depression, suicidal ideation, and substance abuse[23–25]. It is also linked to disruptions in biological processes and to somatic complaints and sleep issues[26,27]. Beyond these negative associations with health, parental burnout is further related to impairments in family functioning and child development. It is linked to more frequent and intense partner conflicts, which heighten marital distress[28], as well as increased likelihood of parental neglect or violence towards children[28,29]. These adverse parenting behaviors can contribute to internalizing and externalizing problems in children, such as anxiety, loneliness, aggression, conduct issues, and difficulties in social interactions[30,31], which ultimately undermine children's overall well-being and mental health[32]. Given the potentially severe consequences of parental burnout, researchers seek to identify its predictors and investigate its underlying mechanisms in depth. Parental burnout is considered a consequence of chronic parenting stress and a lack of resources to cope with or offset parenting demands[1,2,33]. Emotion regulation has gained prominence in understanding the etiology of parental burnout due to its profound function in mental health[4].

Among others, managing the expression of emotions is an integral part of emotion regulation[34]. When such regulation is directed toward achieving interpersonal goals, it is also known as emotional labor[17]. The present study sheds light on genuine expression, a dimension of emotional labor, defined as the appropriate and authentic display of one's true emotions toward others[15,16]. Initially conceptualized in occupational settings, emotional labor encompassed two strategies: surface acting (i.e., suppressing undesired emotions or faking desired emotions) and deep acting (i.e., attempting to change feelings to produce a more genuine display). Subsequent research work expanded this framework to include genuine expression as a third strategy[15,16]. Scholars have argued that emotional labor constitutes a form of emotion regulation aimed at interpersonal goals[35–37] and that this concept can be meaningfully applied to parenting[37,38]. According to Ashforth and Humphrey[39], the heart of emotional labor lies in expressing emotions in accordance with organizational or social norms; thus, even when parents genuinely feel enthusiastic or sad and express these emotions appropriately to their children, they are still engaging in deliberate emotional management. Parental emotional expression is also central to the emotion socialization of children[18]. In this literature, parents' emotional expressions serve specific emotion-related or child-rearing goals, such as relieving parents' own physiological arousal, modeling the acceptability of emotional experience and expression, or guiding children's behavior in particular situations. Morris et al.[18] further suggest that mild and moderate degrees of expression of negative expressions can aid children in learning about emotions and emotion regulation. Consequently, how parents express emotions to their children is not only crucial for adapting to dynamic parent–child interactions but also plays a fundamental role in shaping

children's emotional development and long-term adjustment[18]. Not least, genuine expression is considered resource-conserving, as it fosters congruence between parents' inner-self and outer-behaviors[40] and allows parents to communicate their authentic selves to their children, conveying who they are, what they value and desire, and how they are connected to their children[14]. This perspective aligns with the organismic view of wellness on emotion regulation, which holds that emotion expression enhances well-being and mental health, and strengthens resilience to stress when it supports personal authenticity and autonomous choice[41]. Thus, genuine expression (if habitually used) can be considered an adaptive emotion-related parenting strategy for parents, as it conserves resources and supports authenticity. However, its adaptiveness for children may depend on the parent's ability to express emotions skillfully and sensitively. When parents communicate their genuine emotions in a constructive and developmentally appropriate manner, rather than expressing anger or sadness without reflection or guidance, it can foster positive outcomes for children. In this way, genuine expression, when enacted with skill, serves both parent- and child-focused functions.

Prominent affect theories[34,42,43] suggest that the generation of affect and regulatory efforts are inertial cycles; the implementation of regulation in the affect generation process influences not only the intensity and duration of affective responses but also shapes subsequent affective experiences. Theoretical (e.g., job demands-resources model[44]) and empirical[45,46] insights from related work on job burnout further suggest reciprocity between burnout and emotion regulation. Recently, emotion regulation has been proposed to predict parental burnout after synthesizing 34 primary studies in a meta-analysis[4]. However, most of the studies included in this synthesis were cross-sectional, limiting the ability to draw conclusions about causal relationships. Moreover, this assumption was derived from between-person analyses, which compare different individuals. For example, between-person effects might show that parents who generally report higher use of reappraisal also tend to report lower burnout than other parents. In contrast, within-person effects track changes over time within the same individual, such as whether a parent experiences lower burnout on days when they engage more in reappraisal than they usually do themselves. Methodological debates have pointed out that relationships observed at the between-person level do not necessarily hold at the within-person level[47,48]. As such, these associations may align only under specific conditions regarding the presence, magnitude, and direction of effects[49]. A subsequent longitudinal study by Teuber et al.[8] decomposed variance in parental burnout and emotion regulation (reappraisal and rumination) across both levels of analysis. While their between-person results supported the reciprocal notion, their within-person findings suggested that emotion regulation may be a consequence rather than a cause of parental burnout. Such findings have direct implications for prevention and intervention design and lead to the first main research question of this study: How are parental burnout and genuine expression related to one another within parents?

Affective dynamics are considered adaptive when they enable individuals to respond flexibly to changing environmental demands and internal regulatory processes[11,12]. In our study, we delve into two central yet distinct patterns of within-person affective dynamics—inertia and variability—both of which provide valuable insights into psychological processing and mental health[9–13]. In their seminal work, Kuppens et al.[50] define affective inertia as the resistance of affects to change over time. This resistance often reflects affective rigidity and slowed affective changes, potentially indicating a decoupling of affects from their adaptive regulatory functions. As such, affective inertia is commonly linked to psychological maladjustment[9,12,50]. In comparison, affective variability refers to the magnitude of fluctuations within an individual over time[9,10,13], which is often viewed as functional and supportive of mental health[12,51,52]. However, clinical insights suggest that such fluctuations are not per se adaptive; rather, they signal dysregulation when regulatory processes fail to prevent individuals from exceeding dysfunctional thresholds[9,12,51–53]. This complexity is also seen in therapeutic contexts, where a certain degree of variability in emotional distress across treatment sessions is related to better treatment responsiveness[54], yet

extreme variability, especially in the form of intense and rapid mood shifts, is linked to heightened stress reactivity, poor psychological well-being, and greater risk of affective disorders[11]. Kuppens et al.[50] further posit that individuals can show different combinations of inertia and variability, where inertia reflects the speed of affective changes, and variability reflects the magnitude of those changes. Thus, investigating both inertia and variability offers a fuller picture of affective dynamics. Importantly, however, research on affective dynamics has pointed out that variability in affective states is often confounded with person-specific mean levels, as the two tend to be correlated[6,9]. Consequently, studies examining variability should account for person-specific mean levels to isolate the unique contribution of this dynamic pattern. For simplicity, we use the term "dynamic patterns" to refer to both inertia and variability, while taking into account the confounding effect of the person-specific mean in the present study.

Parental affective experiences are fundamentally dynamic as they depend on internal factors and the specific circumstances of parenting situations[53,55]. Both parental burnout and genuine expression to children are affective experiences specific to the parenting domain and can vary across situations within the same parent. Although parental burnout is traditionally viewed as a chronic syndrome[27], emerging research suggests that its symptoms can vary over time, not only monthly[8] but even on a daily basis[6,56,57]. In one of the few studies examining the dynamics of parental burnout, Blanchard et al.[6] analyzed daily diary data using regression models to assess that inertia, variability, and mean levels of parental burnout sub-dimensions predicted overall burnout severity. They found that inertia of emotional distancing was the most robust positive predictor and interpreted this to mean that parents suffering from more severe burnout are more likely to remain trapped in a state of emotional distance from their children. Notably, studies capturing these dynamics at even finer temporal resolutions remain rare, and it is still unclear whether these findings generalize to shorter timescales. Compared to parental burnout, much less known is about the dynamic patterns of genuine expression, which initially reflects the most functional dimension of emotional labor[14]. While inertia is more commonly studied in negatively valenced affects, more positively valenced affective processes like genuine expression are not as well understood, especially whether these positively valenced processes are (mal)adaptive for psychological adjustment. In our study, variability of genuine expression also refers to within-strategy variability[58], capturing the variation in the usage intensity of this single strategy across situations and time. Findings on whether variability in the use of a specific emotion regulation strategy benefits psychological adjustment are mixed, ranging from negative links[58,59], to non-significant findings[60], and even positive associations[61].

The within-person dynamic patterns of parental burnout and genuine expression are unlikely to be uniform across individuals; rather, they may be shaped by time-invariant sociodemographic characteristics (e.g., gender, socioeconomic status, single parenthood, and the presence of special needs in children) and trait-level psychological factors, such as dispositional emotional expressiveness or susceptibility to burnout. Furthermore, it remains unclear whether such dynamic patterns are linked to the development of burnout and genuine expression in the longer term (also, at the trait level). Understanding the parental factors that influence the magnitude and pattern of these dynamic features can provide deeper insight into which individuals are more vulnerable to maladaptive parenting outcomes. Identifying these associations is not only theoretically important but also has practical relevance for designing personalized interventions[48]. This line of reasoning leads us to our second main research question: Does the extent of within-person dynamic patterns—inertia, variability, and mean levels—of parental burnout and genuine expression vary across individuals, and if so, what are their unique contributions to their longer-term development?

In the present study, we took a within-person lens to investigate the dynamic interplay between parental burnout and genuine expression. Our research objectives were twofold. First, we aimed to examine the temporal, within-person associations between parental burnout and genuine expression during the Christmas festive season in the United Kingdom. Although this time frame may entail additional emotional and logistical demands for

parents, the specific impact of the festive season was not the focus of this study. In Hypothesis 1, we proposed reciprocal relationships between the two constructs, such that higher levels of genuine expression at one moment would predict lower levels of parental burnout at the next moment, and vice versa.

Second, we sought to investigate individual differences in the within-person dynamic patterns of both constructs in the experience sampling period and examine how these patterns relate to their respective baseline and follow-up levels. The dynamic patterns of interest included

- inertia, capturing the carryover of parental burnout and genuine expression from one moment to the next,
- variability, capturing the momentary fluctuations in parental burnout and genuine expression,
- mean levels, representing person-specific average levels of parental burnout and genuine expression across the experience sampling period (included as a potential confounder with variability).

In Hypothesis 2, we expected to find individual differences in all three components of the dynamic patterns for both parental burnout and genuine expression. We further examined whether these components mediated the effects of parents' initial levels of burnout and genuine expression on their respective outcomes at follow-up, while controlling for one another. Given the limited prior research on these dynamic processes in the context of parental burnout, this hypothesis was treated as open and exploratory.

## Methods
### Preregistration and open science
This study was part of a larger, preregistered project investigating parental momentary experiences during the Christmas season. The project[62] was registered on November 16, 2023, and the preregistration protocol is accessible at https://doi.org/10.17605/OSF.IO/4XGA6. Although the specific hypotheses for this study were not explicitly preregistered, they were conceptually aligned with the overarching aims of the broader project. As such, the specific hypotheses tested in this study can be considered exploratory. Data and analysis codes associated with this study are available at https://doi.org/10.17605/OSF.IO/5DNRM. The study protocol (2023/W/021) was approved by the Departmental Review Board of the Department of Occupational, Economic, and Social Psychology of the University of Vienna. Participants provided signed informed consent, and participation was voluntary.

### Participants and procedure
The study was conducted from November 2023 to January 2024 and comprised a baseline survey, a 35-day experience sampling period, and a follow-up survey, respectively, conducted before, during, and after the 2023 Christmas festive season. In November 2023, 380 parents from the United Kingdom, each with at least one child under the age of 10, were recruited via the cloud-sourcing platform *Prolific*. They first completed a baseline survey two weeks before being invited to download the *MindSampler* app on November 30, 2023. This application is designed for collecting intensive longitudinal data and integrates *Qualtrics* surveys.

Participants received three randomly timed daily prompts between 8:00 and 20:00, from November 30, 2023 (Day 1) to January 3, 2024 (Day 35). Of those invited, 315 participants completed at least 30 daily experience sampling surveys throughout the study. On January 15, 2024, 307 of these participants also completed the follow-up survey. Participants received a total compensation of £30, distributed as follows: £4 for completing the baseline survey, £1 for installing the *MindSampler* app, £10 as a flat participation fee for responding to at least 30 notifications during the 35-day experience sampling period, and £5 for completing the follow-up survey. Additionally, participants who completed all three parts of the study received a £10 bonus.

Participants who had completely missing data on the variables of interest at baseline, during the experience sampling period, or at follow-up, as well as those who reported having no children residing in the household,

were excluded from the final dataset. After data cleaning, the final sample consisted of 293 participants, which we treated as representing distinct family units (i.e., no parent dyads), with an average age of 38.16 years ($SD = 6.92$). Participants were asked to indicate their gender (male, female, genderfluid/nonbinary/agender, or do not wish to say). Of the respondents, 188 identified as female and 105 as male. Additionally, 24 participants (8.11%) identified as single parents, while the remaining 269 were married or in a committed relationship. The median annual household net income fell within the range of £50,000–£59,999, which was higher than the UK's median household income of £34,500 in 2023[63]. On average, households had 1.86 children ($SD = 0.83$). A total of 62 parents (21.16%) reported having a child with special educational needs (e.g., ADHD, learning disabilities). No data on race or ethnicity was collected.

### Measures

**Parental burnout at baseline and follow-up**. Parental burnout was assessed at both baseline and follow-up using the parental burnout assessment (PBA)[22], a 23-item measure evaluating four key dimensions: emotional exhaustion (9 items; e.g., "I feel completely run down by my role as a parent"), emotional distancing from children (3 items; e.g., "I do what I'm supposed to do for my children but nothing more"), feelings of being fed up (5 items; e.g., "I can't stand my role as a father/mother anymore"), and contrast with one's previous parental self (6 items; e.g., "I don't think I'm as good a father/mother as I used to be"). Responses were recorded on a 7-point scale (1 = never, 7 = every day). We used the mean values of the scale. Internal consistency for the total scale was high ($\alpha = 0.97$ at both baseline and follow-up).

**Genuine expression at baseline and follow-up**. To capture parents' expressions of genuinely felt emotions toward their children, we adapted the corresponding subscale of the emotional labor scale[15] at both baseline and follow-up. This subscale consisted of three items: (a) "The emotions I express to my child(ren) are genuine", (b) "The emotions I show my child(ren) come naturally", and (c) "The emotions I show my child(ren) match what I spontaneously feel". Responses were given on a seven-point Likert-type scale (1 = strongly disagree, 7 = strongly agree). The adapted version was previously evaluated in a preliminary, unpublished study with a large sample of English-speaking parents, in which the subscale showed sound psychometric properties. We used the mean scores of the scale. Internal consistency for the total scale was high ($\alpha = 0.91$ at baseline, $\alpha = 0.89$ at follow-up).

**Sociodemographic variables**. Sociodemographic variables were measured at baseline and included participants' gender, relationship status, socioeconomic status, and the presence of special educational needs in their children. Gender was dummy-coded (1 = female, 0 = male). Relationship status was assessed by asking participants whether they were single, married, or in a relationship; those who identified as single were categorized as 1 (single parents), while those who were married or in a relationship were classified as 0 (non-single parents). Socioeconomic status was estimated based on participants' total annual net household income, reported on a 13-point scale (1 = <£10,000, 13 = >£150,000). Finally, the presence of special needs was determined by asking parents whether any of their children had special educational needs (1 = yes, 0 = no).

**Momentary parental burnout**. Momentary parental burnout was measured using a four-item scale, adapted from the PBA[22]. Each item represented a distinct subdimension: (a) "I currently feel completely run down by my role as a parent" for emotional exhaustion; (b) "At this moment I don't think I'm the good father/mother that I used to be to my child(ren)" for contrast with previous parental self; (c) "At this moment I can't stand my role as father/mother" for feelings to be fed up; and (d) "Currently I do what I'm supposed to do for my child(ren), but nothing more" for emotional distancing from one's child(ren). Responses were

recorded on a 7-point Likert-type scale (1 = not at all, 7 = extremely). The average individual mean was $M_i = 2.06$ ($SD = 0.99$). The intraclass correlation (ICC) showed that 65.6% of the total variance was attributed to between-person variance. The within-person reliability was $\alpha = 0.79$, and the between-person reliability was $\alpha = 0.86$, calculated following Nezlek[64]. For further details about the method, see the preprint by McNeish[65].

**Momentary genuine expression**. To measure parents' momentary genuine expression to children, we used a single item, "The emotions I am expressing to my children are genuine", adapted from the corresponding subscale of the parental emotional labor scale[15]. Responses were recorded on a 7-point Likert-type scale (1 = not at all, 7 = extremely). The average individual mean was $M_i = 5.78$ ($SD = 0.93$). ICC indicated that 46.9% of the total variance was attributed to between-person variance. Using the measurement error autoregressive method for single-item scales[66], the within-person reliability was 0.70, whereas the between-person reliability was 0.99.

### Data analysis

To test our hypotheses, we employed DSEM[19,20]. It combines structural equation modeling (SEM) with time-series and multilevel modeling (MLM), integrating the strengths of all three approaches to analyze intensive longitudinal data. It adopts the use of latent variables from SEM to decompose observed variables into moment-to-moment fluctuations (within-person components) and trait-level differences (between-person components). Like time-series analysis, it incorporates past values of an outcome and time-varying covariates to predict its current state. Additionally, DSEM accounts for individual differences in these time-series effects, a feature borrowed from MLM. This makes DSEM particularly useful for examining how parental burnout and emotional expression evolve within individuals and interact with one another.

In *Mplus* 8.11[67], we specified our conceptual DSEM model, as illustrated in Fig. 1. The complete model equation is provided in the Supplementary Eq. (1). At the within-person level, we included autoregressive effects ($\varphi_{1i}, \varphi_{2i}$) for both parental burnout and genuine expression. That is, for each individual $i$, parental burnout and genuine expression at time $t$ were regressed on their respective lagged values at $t$–1. These autoregressive effects captured inertia in these experiences during the experience sampling period[6,9]. Cross-lagged effects ($\varphi_{3i}, \varphi_{4i}$) were included to examine potential reciprocal influences: parental burnout at time $t$ was predicted by genuine expression at $t$–1, and genuine expression at time $t$ was predicted by parental burnout at $t$–1. Random effects were included for all these autoregressive and cross-lagged effects. Within DSEM, random effects refer to parameters that are allowed to vary across individuals, capturing between-person differences in the strength or direction of the effects. Furthermore, the residual variances of parental burnout and genuine expression were also modeled as random to allow individuals to differ in the extent of their variability of parental burnout and emotional expression. Note, random residual variances in DSEM are estimated on their log-transformed scale. At the between-person level, we regressed the within-person dynamic parameters—inertia, variability, and person-specific mean levels of both parental burnout and genuine expression—on their respective baseline levels. Throughout this study, we refer to these dynamic patterns as those observed during the experience sampling period. In addition, follow-up levels of parental burnout and genuine expression were regressed on these dynamic parameters. If these relationships held, we further examined potential mediation effects of these dynamic parameters using the MODEL CONSTRAINT function in Mplus (i.e., $a \times b$; baseline $\to^a$ dynamic parameter $\to^b$ follow-up). To account for potential confounds, we included several sociodemographic covariates: parental gender, single parenthood, the presence of a child with special needs, and household net income. These variables were used to predict all variables at the between-person level. Notably, continuous predictors were centered at the between-person level, while binary sociodemographic variables were entered uncentered. In our study, latent

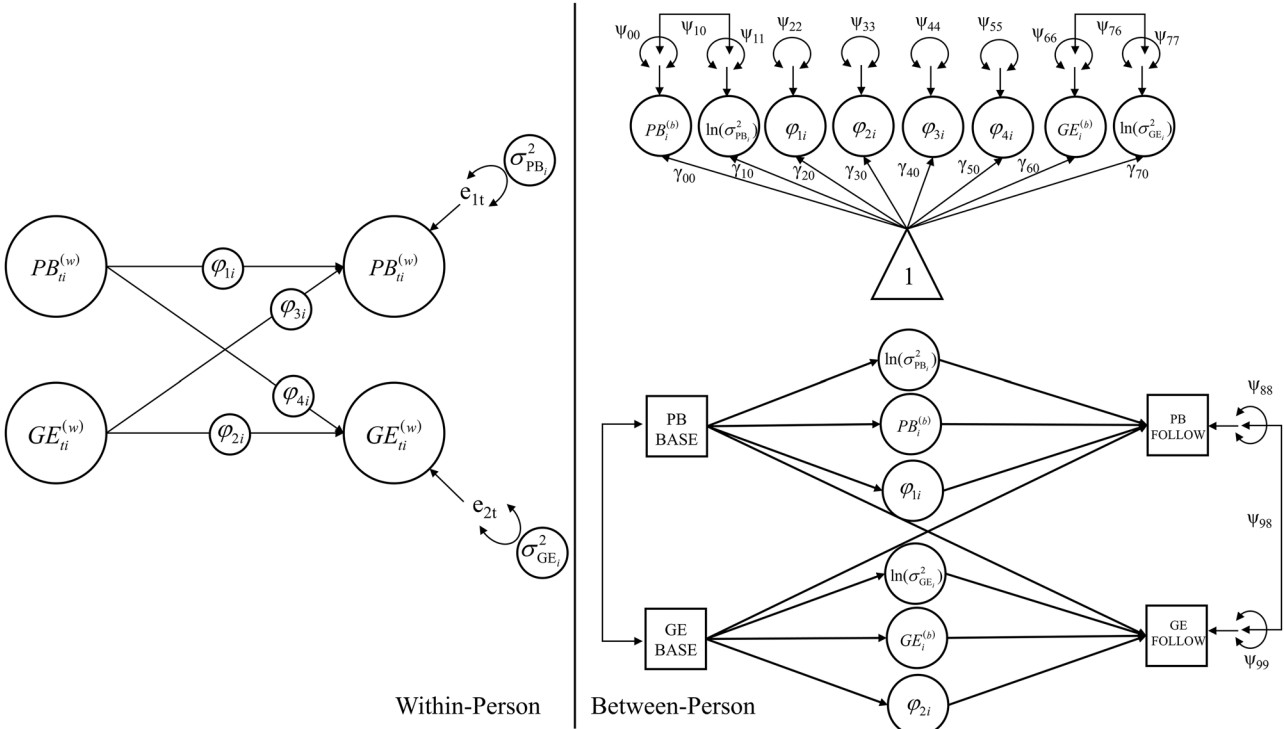

**Fig. 1 | Conceptual model.** *Note.* Sociodemographic variables are not presented here but were statistically controlled at the between-person level by specifying paths from these variables to all other model variables. PB parental burnout, GE genuine expression, BASE baseline measure, FOLLOW follow-up measure.

variable distributions were assumed to be normal, but this was not formally tested.

We conducted DSEM using Bayesian estimation via Markov chain Monte Carlo (MCMC). Model convergence was evaluated using potential scale reduction factors (PSR), which compares variations between chains to variations within chains[68]. PSR values close to 1 indicate good convergence across the MCMC chains, with values below 1.10 generally considered evidence of stochastic convergence[69]. In the final DSEM model, we ran two MCMC chains with 10,000 iterations each and a thinning of 50, resulting in 500,000 total iterations. Default diffuse priors were applied, meaning that the estimates were primarily informed by the data rather than prior assumptions. For each parameter, we report the median of the posterior distribution as the point estimate, along with the corresponding 95% credible interval (CI). Parameters whose CI does not include zero are interpreted as non-null.

### Reporting summary
Further information on research design is available in the Nature Portfolio Reporting Summary linked to this article.

## Results
On average, each of the 293 parents provided 49.32 repeated measures ($SD = 23.06$) during the experience sampling period, yielding a total of 14,451 observations. Table 1 presents descriptive statistics for parental burnout and genuine expression across baseline, the experience sampling period, and follow-up. At the within-person level, after latent decomposition, burnout and expression were moderately correlated ($r = -0.33$ [−0.383, −0.283], $p < 0.001$). Correlations among the variables of interest at the between-person level following latent decomposition are presented in Table 2. Figures 2 and 3 illustrate the trajectories of variability in parental burnout and genuine expression, respectively, over the 35-day experience sampling period, using data from a subsample of 12 participants as illustrative examples. Model convergence for the DSEM was satisfactory, with PSR values for all parameters falling below 1.007 after 500,000 iterations.

**Table1 | Mean (*M*) and standard deviations (*SD*) of parental burnout and genuine expression across the experience sampling period, baseline, and follow-up after latent decomposition using M*plus* (TYPE = TWOLEVEL)**

| | Experience sampling period | | Baseline *M (SD)* | Follow-up *M (SD)* |
|---|---|---|---|---|
| | Between-person *M (SD)* | Within-person *M (SD)* | | |
| Parental burnout | 2.06 (1.00) | 0 (0.72) | 2.70 (1.28) | 2.55 (1.20) |
| Genuine expression | 5.78 (0.93) | 0 (0.99) | 5.76 (1.12) | 5.64 (1.10) |

Latent decomposition was performed only for the experience sampling measures.

### Within-person interplays between parental burnout and genuine expression
The results of DSEM showed that both parental burnout ($B = 0.338$ [0.285, 0.389]) and genuine expression ($B = 0.205$ [0.148, 0.259]) exhibited non-null autoregressive effects over time at the within-person level. This indicated that individuals were likely to report higher levels of parental burnout or genuine expression at one moment if they had experienced elevated levels of the same construct at an earlier moment, relative to their own average. In Hypothesis 1, we expected negative reciprocal relationships between fluctuations in parental burnout and genuine expression at the within-person level. The results partially supported this hypothesis: fluctuations in genuine expression were negatively predicted by prior fluctuations in parental burnout: $B = -0.099$ [−0.132, −0.066]. In other words, when participants experienced higher parental burnout relative to their own average at a given moment, they were likely to report lower genuine expression relative to their own average at the subsequent moment. Contrary to our expectations, however, the reverse cross-lagged effect was likely negligible, with a posterior distribution centered near zero ($B = -0.004$ [−0.015, 0.006]) and substantial probability mass around the null. That is, experiencing higher or lower levels of genuine expression than usual at one moment did not predict subsequent changes in parental burnout relative to one's own average.

**Table 2 | Correlations of variables of interest at the between-person level after latent decomposition using *Mplus* (TYPE = TWOLEVEL)**

| | Variable | 1 | 2 | 3 | 4 | 5 | 6 | 7 | 8 | 9 | 10 |
|---|---|---|---|---|---|---|---|---|---|---|---|
| 1 | Mother | | 0.33 | 0.01 | 0.26 | 0.23 | 0.12 | 0.06 | 0.09 | 0.77 | 0.21 |
| 2 | Needs | 0.06 [−0.06, 0.17] | | 0.00 | 0.01 | 0.41 | 0.06 | 0.55 | 0.75 | 0.07 | 0.25 |
| 3 | Single parenthood | 0.12 [0.03, 0.21] | 0.21 [0.07, 0.35] | | <0.001 | 0.16 | 0.54 | 0.55 | 0.28 | 0.84 | 0.36 |
| 4 | Income | −0.07 [−0.18, 0.95] | −0.14 [−0.26, −0.03] | −0.28 [−0.37, −0.20] | | 0.43 | 0.06 | 0.14 | 0.09 | 0.08 | 0.42 |
| 5 | Burnout baseline | 0.07 [−0.04, 0.18] | 0.05 [−0.07, 0.16] | 0.09 [−0.03, 0.21] | −0.05 [−0.17, 0.07] | | <0.001 | <0.001 | <0.001 | <0.001 | <0.001 |
| 6 | Expression baseline | 0.09 [−0.02, 0.20] | −0.10 [−0.22, 0.01] | −0.04 [−0.16, 0.09] | 0.10 [0.01, 0.21] | −0.39 [−0.52, −0.27] | | <0.001 | <0.001 | <0.001 | <0.001 |
| 7 | Burnout follow-up | 0.10 [−0.01, 0.21] | 0.04 [−0.08, 0.15] | 0.04 [−0.09, 0.16] | −0.08 [−0.19, 0.21] | 0.75 [0.67, 0.82] | −0.36 [−0.49, −0.23] | | <0.001 | <0.001 | <0.001 |
| 8 | Expression follow-up | 0.10 [−0.02, 0.21] | −0.02 [−0.14, 0.10] | 0.06 [−0.05, 0.16] | 0.10 [−0.02, 0.21] | −0.36 [−0.47, −0.24] | 0.47 [0.33, 0.62] | −0.46 [−0.57, −0.35] | | <0.001 | <0.001 |
| 9 | Burnout ESM | −0.02 [−0.14, 0.10] | 0.11 [−0.01, 0.23] | 0.01 [−0.10, 0.12] | −0.12 [−0.26, 0.01] | 0.59 [0.47, 0.71] | −0.28 [−0.41, −0.14] | 0.67 [0.55, 0.78] | −0.39 [−0.53, −0.25] | | <0.001 |
| 10 | Expression ESM | 0.08 [−0.04, 0.19] | −0.07 [−0.19, 0.05] | 0.06 [−0.07, 0.18] | 0.05 [−0.07, 0.17] | −0.40 [−0.52, −0.28] | 0.43 [0.29, 0.57] | −0.50 [−0.60, −0.39] | 0.54 [0.42, 0.65] | −0.50 [−0.62, −0.37] | |

Needs = the presence of special needs in children. ESM = measured during the experience sampling period. Latent decomposition was performed only for the experience sampling measures. Values above the diagonal represent *p*-values, while values below the diagonal indicate correlations along with their 95% confidence intervals.

## Within-person dynamics in parental burnout and genuine expression

In Hypothesis 2, we expected to find individual differences in the within-person dynamic patterns of both parental burnout and genuine expression: inertia, variability, and person-specific mean. Inertia in parental burnout and genuine expression was reflected in their autoregressive effects (with sample averages reported in the previous section), which varied across individuals. This suggested that some participants showed greater inertia than others during the experience sampling period. We also found individual differences in the volatility of burnout and genuine expression, with some participants exhibiting greater variability than others. On average, the within-personal residual variance was 0.179 [0.136, 0.237] for parental burnout and 0.401 [0.284, 0.569] for genuine expression after transforming the estimates to their raw variance scale. These values represent the residual variances when all continuous predictors were centered at zero and binary predictors were set to zero. Finally, mean levels of both constructs also varied across individuals, with the average of 2.022 [1.935, 2.110] for parental burnout and 5.799 [5.705, 5.893] for genuine expression during the experience sampling period.

We further tested the associations between dynamic patterns of parental burnout and genuine expression with their baseline and follow-up levels after controlling for time-invariant sociodemographic variables at the between-person level. The results showed that higher baseline levels of parental burnout predicted greater carryover effect ($B = 0.049$ [0.023, 0.075]), higher person-specific mean levels ($B = 0.507$ [0.427, 0.586]), and greater variability ($B = .675$ [0.532, 0.815]) of parental burnout. After transforming the estimates of variability to their raw variance scale, for each one-unit increase in baseline burnout, the residual variance increased by a factor of 1.964. In practical terms, this means that when a parent's baseline burnout was one unit above the sample average, their variance increased from 0.179 to 0.352. These findings indicated that parents who reported higher burnout at baseline also exhibited systematic differences in their dynamic patterns, characterized by stronger inertia, more pronounced variability, and higher average levels of burnout across the experience sampling period. In contrast, higher baseline levels of genuine expression predicted lower variability ($B = −0.291$ [−0.473, −0.108]) and higher mean levels ($B = 0.363$ [0.276, 0.452]), but did not predict carryover effect ($B = −0.024$ [−0.052, 0.004]). After transforming the estimates of variability to their raw variance scale, each one-unit increase in baseline genuine expression reduced the residual variance by a factor of 0.748. That is, when a parent's baseline genuine expression was one unit higher than the sample average, the variance decreased from 0.401 to 0.300. These findings similarly suggested that parents who reported higher genuine expression at baseline also exhibited systematic differences in their dynamics, marked by higher average levels but reduced volatility in genuine expression. Furthermore, variability and person-specific mean levels were positively correlated for parental burnout ($r = 0.369$ [0.252, 0.507]), whereas for genuine expression, they were negatively correlated ($r = −0.677$ [−0.881, −0.508]). These patterns underscore the importance of controlling for mean levels when modeling the dynamic patterns in relation to baseline and follow-up outcomes.

Linking these dynamic patterns to follow-up levels, we found that follow-up parental burnout was positively predicted by its person-specific mean levels during the experience sampling period ($B = 0.571$ [0.463, 0.681]) but not by its variability ($B = 0.011$ [−0.044, 0.067]) or carryover effect ($B = 0.133$ [−0.294, 0.573]). These effects were observed after accounting for baseline burnout levels ($B = 0.371$ [0.284, 0.458]), baseline genuine expression levels ($B = −0.039$ [−0.107, 0.029]), and socio-demographic covariates (i.e., parental gender, single parenthood, income, and child's special needs). Among these covariates, only parental gender showed a non-null effect ($B = 0.152$ [0.007, 0.299]), suggesting that at follow-up, mothers reported higher levels of burnout compared with fathers. These results indicated that parents who showed higher average levels of parental burnout during the experience sampling period also reported systematically higher burnout at follow-up. In contrast, follow-up levels of genuine expression were positively predicted by its person-specific mean

**Fig. 2 | Trajectories of parental burnout during the experience sampling period in a random subsample of 12 participants.**

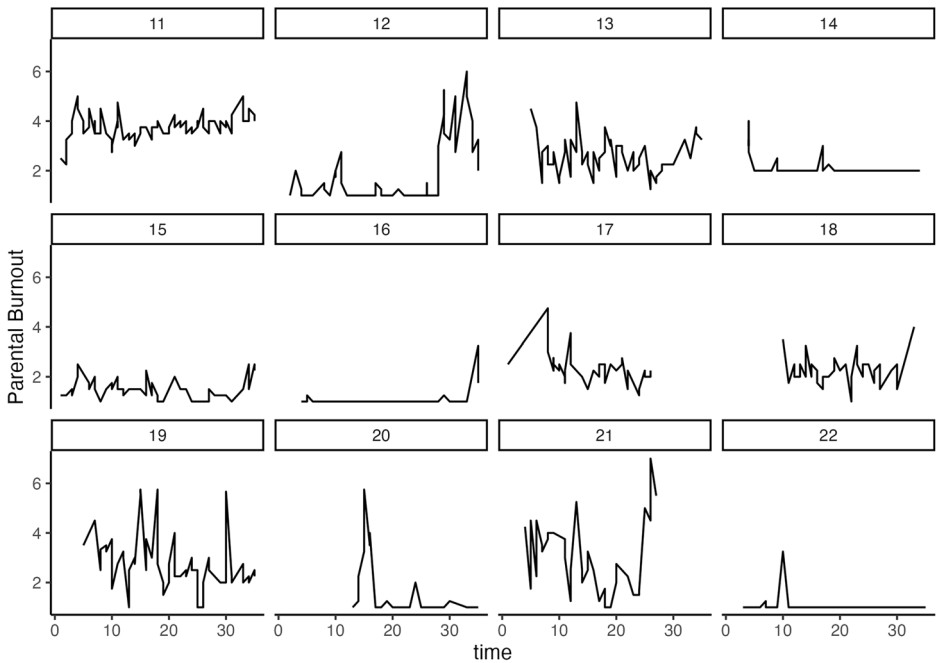

**Fig. 3 | Trajectories of genuine expression during the experience sampling period in a random subsample of 12 participants.**

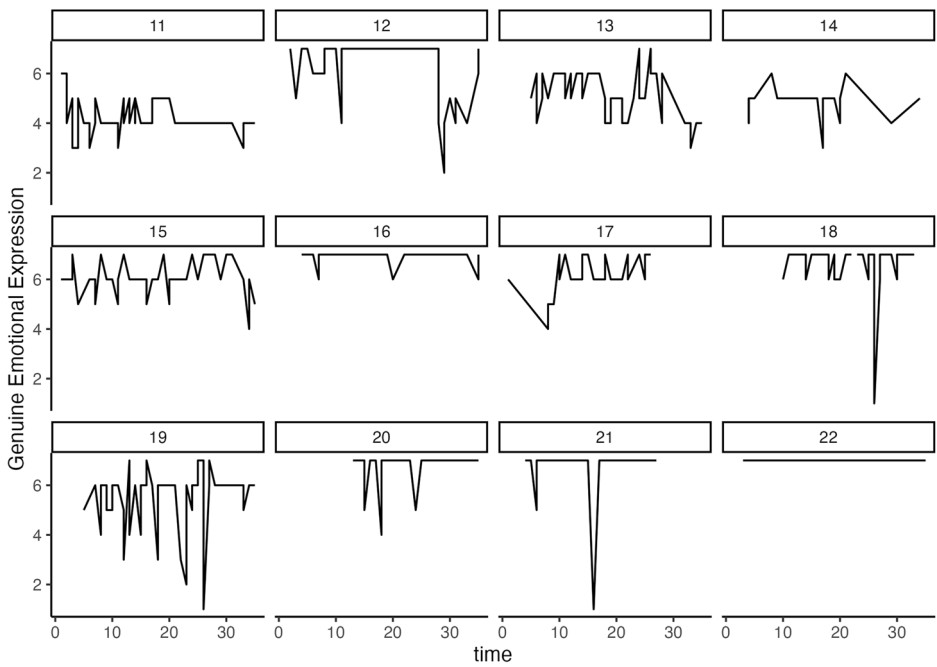

levels during the experience sampling period ($B = 0.233$ [0.127, 0.336]) but not by its variability ($B = -0.034$ [−0.102, 0.031]) or carryover effect ($B = -0.166$ [−0.733, 0.421]), after controlling for baseline expression ($B = 0.515$ [0.354, 0.682]), baseline burnout ($B = -0.066$ [−0.171, 0.039]), and sociodemographics (no evidence; zero included in CI). These results suggested that individuals who showed higher average levels of genuine expression across the experience sampling period reported systematically higher genuine expression at follow-up.

The presence of a direct link between baseline burnout and burnout person-specific mean across the experience sampling period, as well as between this mean and follow-up burnout, pointed to a potential mediating role of person-specific mean levels. Indeed, our analysis confirmed a partial mediating effect ($a \times b = 0.288$ [0.224, 0.365]), indicating that participants who started with higher parental burnout at baseline also demonstrated greater average burnout during the experience sampling period, which, in turn, predicted higher burnout at the follow-up after controlling for relevant covariates. A similar partial mediation pattern was found for genuine expression: person-specific mean levels of genuine expression mediated the relationship between baseline and follow-up genuine expression ($a \times b = 0.168$ [0.118, 0.267]). That is, participants who started with higher levels of genuine expression at baseline also showed higher mean levels during the experience sampling period, which subsequently predicted higher levels of genuine expression at follow-up.

## Discussion

This study took a within-person lens to unpack parental burnout and genuine emotional expression over the emotionally charged Christmas season. We applied DSEM to 35-day intensive longitudinal real-time data,

complemented by baseline and follow-up measures, to investigate whether fluctuations in parental burnout and genuine expression were reciprocally related over time, whether individuals differed in their dynamic patterns (inertia, variability, and mean levels), and whether each of these dynamic patterns was uniquely associated with baseline and follow-up levels of the respective constructs.

### Parental burnout predicted genuine expression at the within-person level

Our first main research objective concerned the relationship between parental burnout and genuine expression at the within-person level. We did not find evidence for reciprocity; rather, DSEM revealed a unidirectional, negative cross-lagged effect, whereby higher-than-usual levels of burnout at one moment predicted lower-than-usual levels of genuine expression at the next moment. This finding suggests that burnout may impair parents' capacity for engaging authentically with their children. When parents feel more emotionally depleted than they typically do (regardless of how they compare to other parents), they may be more likely to suppress or mask their genuine emotions. Instead, they may display emotions they believe as more appropriate or expected in the parenting context (e.g., calmness and patience), even when those emotions are not genuinely felt. This phenomenon reflects emotional dissonance, a form of emotional labor in which there is a discrepancy between felt and expressed emotions[70]. The absence of the reverse spillover effect from genuine expression to burnout suggests that expressing emotions authentically may not be sufficient to interrupt or reverse the cycle of burnout, aligning with Gross's[71] earlier assumption that response-focused strategies such as genuine expression occur relatively late in the emotion-generation process, which significantly reduces their effectiveness in modifying unpleasant experiences. Moreover, this pattern highlights the dominant role of parental burnout in driving emotion regulation. This interpretation is further supported by recent within-person research showing that elevated parental burnout is linked to a reduction in adaptive emotion regulation strategies such as cognitive reappraisal, and an increase in maladaptive strategies like rumination[8]. Together, our findings challenge both the bidirectional assumptions proposed in broader theoretical models and the prevailing unidirectional assumption in the parental burnout literature that emotion regulation predicts parental burnout[4].

### Dynamic patterns of parental burnout and genuine expression

Our second research objective referred to individual differences in within-person dynamic patterns of parental burnout and genuine expression during the experience sampling period—inertia, variability, and person-specific mean—and their associations with baseline and follow-up levels of these constructs. In line with our hypothesis, we uncovered individual differences in all these dynamic patterns. Notably, higher baseline burnout predicted greater inertia, increased variability, and higher person-specific mean during the experience sampling period. As noted by Kuppens et al.[50], individuals can simultaneously exhibit both high inertia and high variability, where inertia reflects the speed of affective changes, and variability reflects the magnitude of changes. Accordingly, our results suggest that participants with higher trait parental burnout show less sudden changes in their burnout symptoms, but experience these symptom changes more intensely. This pattern may point to diminished or slowed responsiveness to changed situational demands, with large fluctuations in burnout symptoms likely reflecting dysfunctional affective processing. From the theoretical perspective, these findings are consistent with the nature of parental burnout as a stress-related syndrome underpinned by exhaustion, detachment, shame, and loss of joy and parental agency[22,27]. The heightened variability likely reflects fluctuating, situation-specific affective responses characteristic of parental burnout, such as cycles of intense emotional reactivity followed by attempts to regain control, and subsequent feelings of frustration or helplessness when those efforts fail.

Interestingly, however, only the person-specific mean, and not inertia or variability across the experience sampling period, mediated the association between baseline and follow-up parental burnout. This finding diverges from results reported in a daily diary study by Blanchard et al.[6], who found that inertia was a more robust predictor of burnout severity than variability or mean. A likely explanation for this discrepancy lies in differences in study design and timescale. Whereas Blanchard et al.[6] used once-daily retrospective assessments over a period of three or eight weeks, our study captured moment-to-moment affective dynamics across a 35-day emotionally intensified period (i.e., the Christmas season) using multiple assessments per day. Nevertheless, our findings align with previous work on the inertia-variability paradox in the context of negative affect predicting depression[9], which suggests that dynamic patterns like inertia and variability can be elevated in clinical samples but not directly predict long-term symptom development once person-specific mean levels are accounted for. Importantly, our findings do not diminish the relevance of these dynamic patterns. On the contrary, we observed that higher baseline parental burnout was associated with greater variability and stronger inertia, suggesting that trait-like burnout co-occurs with more inflexible, slower adaptation to situations, along with larger changes in burnout experiences in daily life. These dynamics may jointly function as proximal risk indicators or early warning signals for persistent burnout. However, the finding that only person-specific mean levels predicted follow-up burnout suggests that these averages capture accumulated emotional experiences and provide insights into temporal dynamics[11]. This makes mean levels a more robust predictor of the development of parental burnout.

Turning to genuine expression, our results showed that higher baseline levels of genuine expression predicted smaller variability and higher person-specific mean levels across the experience sampling period, but not inertia. This pattern suggests that parents who are generally more capable of expressing their emotions genuinely are not only more likely to do so consistently (high mean) but also less likely to experience erratic fluctuations in expression (low variability). It may signal healthy emotion regulation, wherein individuals with higher trait genuine expression can respond to environmental changes without experiencing large or erratic fluctuations in their emotional displays. The lack of associations between baseline expression and inertia is also noteworthy. While affective inertia has been linked to maladaptive processes such as rumination, emotional rigidity, and psychological maladjustment[12,50], the absence of a link here may indicate that inertia is less pronounced for understanding the dynamics of emotional processes like genuine expression (of any emotions). Unlike rigid affective patterns, genuine expression in the framework of emotional labor involves open yet context-appropriate communication of emotions, which likely requires adaptability and flexibility on the part of parents. Indeed, prior work has suggested that the functional significance of inertia might differ depending on the affective valence and context[11]. Similar to parental burnout, only person-specific mean levels of genuine expression during the experience sampling period served as a mediating role between its baseline and follow-up levels. This finding suggests that, beyond short-term affective fluctuations, it is the cumulative or average tendency to express emotions authentically that carries forward into longer-term emotional expression. It also resonates with research on emotional authenticity, which highlights the importance of sustained, congruent emotional expression for well-being and relationship quality[40,41]. The fact that variability in genuine expression did not serve as a mediator may indicate that short-term fluctuations in emotional authenticity are less influential, or possibly more context-dependent, when it comes to shaping longer-term adjustment. Instead, consistently high levels of genuine expression may reflect an adaptive emotional orientation that supports psychological resilience over time.

### Empirical and practical implications

Most research on parental burnout and emotion regulation has relied on cross-sectional data, between-person analyses, and retrospective self-report measures. Such approaches are susceptible to memory biases and schematic distortions, as individuals tend to report how they believe they usually feel, rather than how they felt in a specific moment[72,73]. Laboratory-based

assessment, although valuable for examining causality, often suffers from the limitation of transferability to natural parenting settings. To address these limitations, the current study employed ESM to capture parents' affective experiences in real time and in their natural environments. This approach minimizes memory biases and enhances the ecological validity of affective assessments[74,75]. In affective science, dynamics have long been of interest. However, most studies have quantified variability using within-person standard deviation or coefficients of variation across repeated measures[58], modeled inertia and variability in MLM[76] or regression models, and mainly looked at zero-order associations between these dynamic patterns and affective outcomes[11]. We applied DSEM to decompose observed variables into between- and within-person components and modeled dynamic patterns as latent constructs. Compared to DSEM, both MLM and person-level variability indices have notable limitations. MLM cannot capture temporal dynamics and often assumes independent errors across time, limiting its ability to model how parental burnout and genuine expression influence each other over time. Person-level variability metrics further reduce rich time-series data into static summaries, losing information about timing and directionality. Neither approach adequately models individual differences in dynamic patterns nor accounts for measurement error. In contrast, DSEM combines time-series modeling, MLM, and SEM, allowing researchers to examine within-person dynamics and between-person differences in parental burnout and genuine expression over time, while handling missing data and uneven time intervals more robustly[19,20].

Our findings carry important practical implications for interventions and parental support strategies. Parental burnout is a globally prevalent phenomenon, with growing evidence suggesting it can affect parents across diverse cultural, socioeconomic, and family contexts[1,2]. The finding that burnout impairs parents' ability to express emotions genuinely has implications for parent-child relationships. Parental emotional authenticity plays a key role in the emotion socialization of children, as well as in fostering secure, positive parent-child relationships and well-being of both parents and children[18,41]. It is key to identify and intervene in parental burnout at an early stage. However, in the context of burnout, emotional authenticity becomes more complex. Parents experiencing intense burnout-related emotions (i.e., emotional exhaustion, despair, detachment, or the desire to withdraw from parenting) may consciously suppress these feelings to protect their children and maintain family functioning. In such cases, reduced emotional authenticity may serve as a short-term adaptive coping strategy rather than indicating an emotional deficit. Despite its potential short-term utility, the prolonged suppression of genuine emotions can lead to significant long-term consequences[77,78]. On the one hand, concealing one's true emotional state may increase the psychological burden on parents and undermine their well-being. On the other hand, parents may eventually reach a breaking point, resulting in sudden emotional outbursts or, in extreme cases, harmful behaviors toward their children. Thus, a temporary reduction in emotional authenticity may initially serve a protective function, but it ultimately underscores the importance of early detection and intervention to prevent maladaptive escalation. One promising direction involves mindfulness-based parenting interventions[79], which aim to enhance parents' emotional awareness and reduce automatic reactivity. These interventions promote present-moment attention and nonjudgmental acceptance of difficult emotions, which may not only buffer against burnout but also support genuine emotional expression. Other approaches, such as emotion coaching or stress inoculation training, could also be integrated into broader parental support programs. The timing of our study (i.e., during the emotionally heightened Christmas season) further underscores the need to consider contextual emotional stressors in tailoring interventions. Holidays can amplify the emotional load on parents, particularly when navigating family obligations, financial strain, and elevated expectations for caregiving. Intervention programs should therefore incorporate seasonal and situational stressors into their assessment and planning, recognizing that parental burnout is not static but dynamically shaped by environmental demands. Finally, ESM-based feedback[80,81] may enhance personalized parenting support. For instance, if a parent shows

high fluctuations and a high average level in parental burnout, an intervention can be tailored to provide targeted support in those specific moments through reminders, micro-interventions, or reflective prompts delivered via mobile applications.

## Limitations

Several limitations of this study have to be acknowledged, which may also open avenues for future research. First and foremost, although genuine expression can theoretically be considered a parenting strategy related to emotion regulation, the measure used in this study did not capture several important dimensions: the valence of the expressed emotion (i.e., whether it was positive or negative), the context sensitivity of expression (e.g., whether emotions were expressed in a reflective and appropriate manner), or the explicit goals underlying these expressions (e.g., signaling that experiencing emotions is acceptable). This is a notable limitation, particularly given research showing that frequent expression of negative emotions by parents can be associated with adverse outcomes for children[82]. To advance understanding of genuine expression as a regulatory strategy in parenting, future research should differentiate between the types of emotions expressed, the sensitivity of their expression, and the underlying goals. Second, the results relied on self-reports, and the possibility of bias due to method effects cannot be ruled out. While self-reports are deemed the most appropriate way to gain insights into parents' internal world[83] within the ESM framework, future studies could benefit from incorporating multi-method approaches, such as physiological markers (e.g., heart rate variability) for affective functioning[84] or reports from partners in a dyadic ESM setting[85] to gain a more comprehensive understanding of parental affective functioning. Third, our sample consisted of parents from the UK during a specific cultural season (Christmas), which limits the generalizability of the findings to broader, more diverse parent populations. Fourth, the intensive longitudinal design required high participant engagement, which may have resulted in a sample of parents with better self-regulatory skills than the average population. This introduces potential selection bias, as parents with lower self-regulation or coping skills might be less likely to participate. Fifth, how parents express their emotions and perceive their parenting role may vary across cultures[86]. On average, participants in our study reported relatively high levels of genuine expression. However, different patterns may emerge in Eastern cultural contexts, where norms around emotional expression can vary significantly. To enhance the generalizability of the findings, future research should aim to recruit a more diverse sample of parents, including those from varied cultural and sociodemographic backgrounds. Finally, researchers in clinical psychology have emphasized that distinguishing between- and within-person associations is essential for understanding psychopathology and advancing science–practice integration[48]. Building on this insight, we call for greater research efforts using intensive longitudinal designs and DSEM. Such work can provide a more nuanced and actionable understanding of parental burnout, ultimately informing more effective and personalized intervention strategies.

## Conclusion

This study contributes to the literature by providing in-depth insights into the dynamic affective experiences of parents during the Christmas festive season. Through a within-person lens and state-of-the-art methodology, we captured real-time dynamic patterns of, and associations between, parental burnout and genuine emotional expression in parents' natural environments. Our findings revealed a negative, unidirectional association from parental burnout to genuine expression. Furthermore, parents with higher baseline burnout experienced greater inertia and variability in burnout across the festive season, along with higher mean levels. In contrast, higher baseline genuine expression was associated with fewer fluctuations and consistently higher levels of genuine expression. Crucially, only person-specific mean levels of both constructs served as mediators between baseline and follow-up assessments, underscoring the central role of cumulative emotional experience over time in shaping longer-term outcomes. Future research should continue to explore these dynamics across other emotionally demanding periods and within more

diverse family contexts to better inform prevention and intervention efforts.

## Data availability
Data associated with this study are available at https://doi.org/10.17605/OSF.IO/5DNRM.

## Code availability
Analysis codes associated with this study are available at https://doi.org/10.17605/OSF.IO/5DNRM.

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

## Author contributions

Z.T.: Conceptualization, methodology, formal analysis, data curation, writing—original draft, writing—review and editing, visualization, project administration. E.B.: Conceptualization, data curation, writing—review and editing, project administration. J.R.: Conceptualization, data curation, writing—review and editing, project administration. S.G.: Writing—review and editing. K.A.: Writing—review and editing. D.M.: Methodology, formal analysis, writing—review and editing, visualization, and supervision.

## Competing interests

The authors declare no competing interests.
