## [Transparent Peer Review file · Communications Psychology]

Higher momentary parental burnout predicts lower subsequent emotional expression in parents during the festive season

Corresponding Author: Dr Ziwen Teuber

Version 0:

Decision Letter:

Dear Dr Teuber,

Thank you for your patience during the peer-review process. Your manuscript titled "Unwrapping the Dynamics of Parental Burnout and Genuine Expression During the Festive Season: A Dynamic Structural Equation Modeling Approach" has now been seen by 2 reviewers, and I include their comments at the end of this message. They find your work of interest but raised some important points. We are interested in the possibility of publishing your study in Communications Psychology, but would like to consider your responses to these concerns and assess a revised manuscript before we make a final decision on publication.

We therefore invite you to revise and resubmit your manuscript, along with a point-by-point response to the reviewers. Please highlight all changes in the manuscript text file.

Editorially, we consider it especially important that the revision addresses the reviewers' concerns regarding the conceptualization of genuine emotional expression as an emotion regulation strategy. As per the reviewer's suggestions, it may help to connect this construct more clearly to well-established theoretical models of emotion regulation. Furthermore, both reviewers had concerns regarding the framing of the paper and the literature used as justification for examining this construct. Please avoid causal claims derived from correlational evidence. Please ensure data and code (ideally, annotated) are available for reviewers upon resubmission.

I am attaching an Editorial Requests Table that details critical reporting requirements for the revised manuscript. Please attend to each item and ensure your manuscript is fully compliant. If your revised manuscript is not aligned with these requests on major issues, such as those concerning statistics, it may be returned to you for further revisions without re-review.

Please submit the following items:

- Revised manuscript
- Point-by-point response to the referees' comments
- Cover letter (as a separate document)
- [Nature Research Reporting Summary](https://www.nature.com/documents/nr-reporting-summary.pdf)
- Completed Editorial Request Table (attached).

via this link: Link Redacted .

Additional guidance is available in our style and formatting guide Communications Psychology formatting guide.

Best regards,

Lameese Eldesouky

Lameese Eldesouky, PhD
Editorial Board Member
Communications Psychology
orcid.org/0000-0003-4977-8203

REVIEWER EXPERTISE:

Reviewer #1 parenting, emotion regulation
Reviewer #2 emotion regulation, experience sampling

REVIEWER REPORTS:

Reviewer #1 (Remarks to the Author):

Thank you for the opportunity to review this manuscript, titled "Unwrapping the Dynamics of Parental Burnout and Genuine Expression During the Festive Season: A Dynamic Structural Equation Modeling Approach." The study design is strong given its use of dynamic structural equation modeling to analyze intensive longitudinal data. However, I have concerns about construct conceptualization and the introduction that limit my enthusiasm about the manuscript in its current form. Below I offer some suggestions that I hope will help this work move forward.

Major comments:

I have concerns with authors' characterization of genuine emotional expression as a form of emotion regulation. Authors introduce genuine emotional expression as a specific form of emotion regulation in parenting on p. 1, but do not provide citations for this claim. In reviews of parental emotion regulation (Zhang et al., 2023 <https://doi.org/10.1016/j.dr.2023.101092>; Zimmer-Gembeck et al., 2021 <https://doi.org/10.1177/01650254211051086>) and scales assessing parental emotion regulation (Pereira et al., 2017 <https://doi.org/10.1007/s10826-017-0847-9>; Rodriguez & Shaffer, 2021 <http://doi.org/10.1037/fam0000808>), genuine emotional expression is not mentioned. Further, in models of parental emotion socialization by Nancy Eisenberg and Amanda Morris (see also Hajal & Paley, 2020 <https://doi.org/10.1037/dev0000864>; Edler & Valentino, 2024 <https://doi.org/10.1037/bul0000423>), parental emotional expressiveness is conceptualized as a behavior distinct from parental emotion regulation. In a revision, authors could remove mentions of genuine emotional expression as a form of emotion regulation. Indeed, the cited Yang et al. (2019) study was focused on emotional labor in the workplace; I do not believe that Yang and colleagues suggested that genuine emotional expression was a form of emotion regulation.

Authors may have a different perspective and are welcome to present it in their manuscript. That said, the proposal that genuine emotional expression is a form of emotion regulation should be supported by a more extensive argument and citations. Or, the claim could be stated explicitly as a novel proposal. Another idea is for authors to conceptualize their construct as a lack of expressive suppression. This would cohere with the literature on parental emotion regulation, James Gross' key process model of emotion regulation, and the meta-analysis by Brandão et al. (2024) that authors cite.

Regardless of whether authors continue to (1) refer to their construct as genuine emotional expression and (2) claim it is a form of emotion regulation, they also must contend with the limitation that the items do not capture the valence or extremity of emotional expressions (in contrast, Halberstadt's oft-used 1995 Self-Expressiveness in the Family Questionnaire <https://doi.org/10.1037/1040-3590.7.1.93> distinguishes between positive and negative emotional expressiveness). On p. 5 authors state that "genuine expression can be considered an adaptive emotion regulation strategy if habitually used." What if parents are expressing negative emotions? There are a number of studies connecting parents' frequent/high levels of negative emotional expressiveness to poor outcomes for youth (see meta-analysis by Zinsser et al 2021 <https://doi.org/10.1016/j.ecresq.2021.02.001>).

The remainder of my comments are organized by section.

Abstract:

Authors write "Dynamic structural equation modeling (DSEM) was used to test reciprocal within-person relations between both constructs over time, to assess individual differences in dynamic patterns, and to explore whether these patterns played a mediating role in changes from baseline to follow-up." Authors should state the constructs in which changes from baseline to follow-up are considered.

Authors state that findings "highlight the importance of personalized, real-time interventions that target emotional regulation and burnout recovery in parents ..." This could be taken to imply that the present study analyzed an intervention; rewording can clarify that the study has implications for the design of intervention programs.

Introduction:

On p. 3, authors use Brandão et al. (2024) as a citation for "the profound influences of emotion regulation on mental health". The opening to the manuscript would be even stronger if authors instead reference Brandão for their meta-analysis of the association between parental burnout and parental emotion regulation (Brandão's analyses do not examine parental burnout and parental mental health associations). Similarly, authors write on p. 4 that parental burnout is linked to "increased likelihood of parental neglect or violence towards children (Brianda et al., 2020; Kawamoto et al., 2018)"; Kawamoto et al. (2018) did not examine parental neglect or violence as an outcome.

On p. 3, authors write: "However, the directionality of these variables remains empirically inconsistent..."; authors likely mean the directionality of effect between these variables remains empirically inconsistent.

On p. 4, authors use causal language in the following sentences: "If left untreated, parental burnout can lead to more severe mental issues, including depression, suicidal ideation, and substance abuse (Brianda et al., 2020; Kawamoto et al., 2018). It also disrupts biological processes and contributes to somatic complaints and sleep issues (Mikolajczak et al., 2019). Beyond these negative influences on health, parental burnout impairs family functioning and child development." In the case that any of these studies were nonexperimental, causality should not be implied.

A notable strength of this study is its ability to disaggregate between within- and between-person effects. On p. 6, authors do a nice job describing that prior studies on parental burnout and parental emotion regulation have generally not been longitudinal or used methods that disaggregate these between/within effects. For readers less familiar with the distinction between within- and between-person effects, authors could provide an example of each.

On p. 7 authors write that "Affective dynamics are generally considered adaptive"—this is a broad statement that authors contradict when they go on to write "affective inertia is commonly linked to psychological maladjustment"; rewording will fix this.

On p. 8 authors describe the interesting findings from Blanchard et al. (2025) and say that "They found that inertia was the most robust predictor." Can authors describe this finding in more detail? Was inertia in parental burnout positively associated with overall burnout severity (or negatively)?

On p. 8-9 authors state that genuine expression "initially reflects the positive side of emotional labor (Humphrey et al., 2015)." It is not clear what authors mean, perhaps they should define emotional labor. The following sentence "While inertia is more commonly studied in negatively valenced affects, more positively valenced affective processes like genuine expression are not as well understood, especially whether these positively valenced processes are (mal)adaptive for psychological adjustment." also could be clearer.

When authors write on p. 9 that "Findings on whether variability benefits psychological adjustment are mixed, ranging from negative links (Blanke et al., 2020; Elkjær et al., 2022), to non-significant findings (Wang et al., 2021), and even positive associations (Aldao & Nolen-Hoeksema, 2013)", it seems relevant what the construct is. Variability in parental burnout? Variability in genuine emotional expression? Variability in and of itself is not meaningful without knowing the construct. I tried to look up these papers but none are in the reference section. I noticed Lin et al. (2023) and English & John (2013) are also missing from the reference section.

Current Study/Method/Data Analysis/Results:

These sections were well-written. Authors do a good job explaining the dynamics of interest (inertia, variability, person-specific means) and DSEM. Their writeup of the results was overall clear. I only have a few suggestions.

Authors should describe what a random effect is in the context of a DSEM for less familiar readers.

As DSEM is still a relatively new method, I wonder if authors could include their equations and a figure representation of the model (as in Hamaker et al., 2018 <https://doi.org/10.1080/00273171.2018.1446819> and McNeish & Hamaker, 2020 <http://doi.org/10.1037/met0000250>).

On p. 19, authors state that “parental gender showed a non-null effect ($B = .152 [.007, .299]$)” on follow-up parental burnout. Back on p. 13 authors state how gender was dummy-coded, but it should be restated in the results or reworded to indicate that mothers vs. fathers were reporting more burnout at the follow-up assessment.

Discussion:

Clear interpretation of the within-person cross-lagged effect.

Good explanation of how both inertia and variability can be elevated.

A key limitation to note in the limitations/future directions section is that the items for genuine emotional expression do not capture the valence or extremity of emotional expressions.

Reviewer #2 (Remarks to the Author):

Unwrapping the Dynamics of Parental Burnout and Genuine Expression During the Festive Season: A Dynamic Structural Equation Modeling Approach.

SUMMARY: The Authors investigate the relationship between parental burnout and genuine expression during the Christmas period in a large sample of parents in the UK, using a pre-existing ESM dataset. The authors focused on two key research questions: 1. Is there a reciprocal relationship between parental burnout and genuine expression (within-person). 2. Do individual differences in within person patterns (inertia, variability and person mean) of parental burnout and genuine expression mediate their association from baseline to follow-up.

I found the research question worthy of exploring (although I am confused by some of the literature used to justify the investigation), and the introduction is generally well written, expressing clear arguments. The method matches the research question and is generally clearly explained. A strong point of the paper is the clarity with which the authors describe DSEM — very well done there. Yet, I have some comments concerning OSF practices, results and interpretations. I hope the authors will take the comments as opportunities to strengthen their work, and I want to highlight that I genuinely enjoyed reading their paper.

1. Major points

- a. I am confused by the framing of genuine expression as an “emotion regulation strategy” (line 92) in the context of this specific paper. While the points made on page 5 are understood and potentially valid, to my understanding of the Process Model of ER (Gross, 1998; 2015), Affect/Emotion Regulation strategies are implemented to increase, decrease, or maintain emotions. This aspect seems to be mostly missing from the conceptualization offered in the introduction, and no reference is made to the IER literature—this literature would have been perhaps more directly relevant, considering that the Authors seems to allude to the fact that in the parenting context, emotion expression is fundamental in shaping children’s emotional socialization and long-term development (line 82-84). My confusion is also matched by the methodological operationalization of the “emotion regulation strategy”. When looking at the phrasing of the item (line 287-294), there is no indication of this variable being “a strategy” (i.e., used for the intent of regulating one’s emotions, or their children’s emotion).
- b. Could authors please report correlations (within and between person) for the variables of interest (in the manuscript, or in the supplementary materials). Did the authors check for multicollinearity?
- c. The data collection of this project was pre-registered, however there is no reference in the pre-registration speaking for the “conceptual alignment of the hypotheses with the overarching aims of the broader project” (line 210-212). The variables and hypotheses were not explicitly stated in the pre-registration. The code available on the OSF is difficult to navigate, even more so without the data available to reproduce the analysis. Perhaps authors could knit a document and upload to the OSF if data cannot be made available on initial submission? No information on the data preprocessing and the decision-making concerning rationale for exclusions is provided. This is a problem stemming from the quality of the data collection pre-registration, and flowing on to the level of information provided in the manuscript (and supplementary materials). There was no rationale for data collection sample size (e.g. power). Unfortunately, I cannot consider this paper to be aligned with best open science practices.
- d. Line 410-420: I do apologize if I somehow missed it, but I am unclear on how exactly the mediation analyses were set up. Then, in the results section, it is not specified if the mediation is partial or full. A figure, and/or a table of results would maybe make it clearer?
- e. Line 561-582: Given my other comments about theoretical points, and operationalization of the main variables, I think the recommendations over-reach with interpreting the findings. I recommend the authors remove this part, or significantly tone

down the recommendations.

2. Minor Points

- a. The paper seems to put a major weight on the fact that data was collected during the festive season. The justification for this point is clear, and is understandable, however, I note that there was no real measure of the impact of the festive season. Accordingly, I think this should be a point that is clearly made in the method, but not leveraged as much in the title, and discussion, to frame the results as if there was a measured role of the festive season, on the relationship between variables.
- b. Line 95: Gross 1998b, 2025, 2024 are cited, but are missing from the reference list.
- c. Line 105: perhaps another more recent and useful reference for the point the authors make is: Hopwood et al (2025). I think this might also be useful on line 176-177, and in the discussion, if the authors wanted to keep a point about the merits of DSEM in disentangling within from between person effects, and perhaps indicate future directions along the same lines (but as I stated in one of my other points, toning down the claims currently made in the discussion).
- d. It would have been useful to have an idea of the amount of sample stemming from the same parental dyad.
- e. The language used in the results section is sometimes confusing as to what exact analysis have been conducted. E.g. line 350: "more likely to report..." Does this mean that analyses were logistic, binarizing the variable (which to my understanding was on a 1-7 scale)? Similar issue with line 357 "they tended to...". Line 360 "did not lead to subsequent changes..." this is causal language, which is incorrect even when using lagged analyses or DSEM, as the data is still correlational. Line 496-497 "more likely..... less likely..." again, logistic analyses?
- f. Line 504: "... inertia is less pronounced for understanding the dynamics of positively valenced emotional processes like genuine expression": this statement makes an assumption that people would be only expressing positive emotions genuinely.
- g. Line 556-561: please provide citations and references for the statements made.

REFERENCES

Hopwood, C. J., Aafjes-van Doorn, K., Békés, V., Luo, X., Ringwald, W. R., & Wright, A. G. C. (2025). Is psychological research producing the kind of knowledge clinicians find useful? *American Psychologist*. Advance online publication. <https://doi.org/10.1037/amp0001538>

* TRANSPARENT PEER REVIEW: Communications Psychology uses a transparent peer review system. This means that we publish the editorial decision letters including Reviewers' comments to the authors and the author rebuttal letters online as a supplementary peer review file. However, on author request, confidential information and data can be removed from the published reviewer reports and rebuttal letters prior to publication. If your manuscript has been previously reviewed at another journal, those Reviewers' comments would not form part of the published peer review file.

Version 1:

Decision Letter:

Dear Dr Teuber,

Your manuscript titled "Unwrapping the Dynamics of Parental Burnout and Genuine Expression: A Dynamic Structural Equation Modeling Approach" has now been seen by our reviewers, whose comments appear below. In light of their advice I

am delighted to say that we are happy, in principle, to publish a suitably revised version in Communications Psychology.

We therefore invite you to revise your paper one last time to address the remaining concerns of our reviewers and a list of editorial requests. At the same time we ask that you edit your manuscript to comply with our format requirements and to maximise the accessibility and therefore the impact of your work.

EDITORIAL REQUESTS:

SUBMISSION INFORMATION:

OPEN ACCESS:

* DATA AVAILABILITY:

Link Redacted

Best regards,

Jennifer Bellingtier

Jennifer Bellingtier, PhD
Senior Editor
Communications Psychology

REVIEWER EXPERTISE:

Reviewer #1 parenting, emotion regulation

Reviewer #2 emotion regulation, experience sampling

REVIEWERS' COMMENTS:

Reviewer #1 (Remarks to the Author):

I commend the authors on their comprehensive and responsive revised manuscript. The revisions have strengthened the manuscript, and I think it makes an important contribution to the literature. I only have a few minor comments/suggestions:

1. In the abstract, might authors add the valence of the effect to the sentence: "Results revealed a unidirectional within-person association from parental burnout to genuine expression"
2. There seems to be an inconsistency on page 13. Authors write, "315 participants completed at least 30 daily experience sampling surveys throughout the study," then go on to say "A total of 247 participants responded to more than 30 prompts".
3. On page 29 authors write, "However, the fact that only person-specific mean levels predicted follow-up burnout suggests that mean levels of burnout reflect accumulated emotional experiences and may entail information on temporal dynamics (Houben et al., 2015), thereby playing a more robust predictor of development of parental burnout." The grammar at end of this sentence is confusing and could use minor revision.

Reviewer #2 (Remarks to the Author):

I greatly appreciate the authors' thorough responses to the reviews and believe that this revision has done well in strengthening the manuscript. I only have one comment (not requiring any change to the manuscript), as I believe all of my concerns have been sufficiently addressed.

In reference to my comment on open science practices:

I appreciate the changes made to the manuscript to highlight how the hypotheses were not explicitly pre-registered. Indeed, as the Authors mention in their rebuttal letter, it can be difficult to pre-register all possible hypotheses when collecting data for multiple projects (particularly in the case of very rich datasets such as ESM/EMA/AA data). It is common practice, in case of use of pre-existing datasets, to submit a pre-registration of analyses of pre-existing data (sometimes referred to as post-registration), where it is possible to delineate hypothesis for the new project, and being transparent on Authors' preexisting knowledge of the dataset. A template for this type of registrations is available on the OSF (van den Akker et al., 2021), and usefulness of this approach is remarked in a few papers (e.g., Mertens & Krypotos, 2019).

Mertens, G., & Krypotos, A. M. (2019). Preregistration of Analyses of Preexisting Data. *Psychologica Belgica*, 59(1), 338–352. <https://doi.org/10.5334/pb.493>

Van den Akker, O. R., Weston, S., Campbell, L., Chopik, B., Damian, R., Davis-Kean, P., Hall, A., Kosie, J., Kruse, E., Olsen, J., Ritchie, S., Valentine, K., Van 't Veer, A., & Bakker, M. (2021). Preregistration of secondary data analysis: A template and tutorial. *Meta Psychology*, 5, MP.2020.2625. <https://doi.org/10.15626/MP.2020.2625>

Again, I appreciate the responsiveness of the authors to my comments, and genuinely enjoyed reading their work and reflecting on their findings. Great work!

Reviewer Comments, Author Responses, and Manuscript Changes

Reviewer #1

1. **Reviewer's comment:** Thank you for the opportunity to review this manuscript, titled "Unwrapping the Dynamics of Parental Burnout and Genuine Expression During the Festive Season: A Dynamic Structural Equation Modeling Approach." The study design is strong given its use of dynamic structural equation modeling to analyze intensive longitudinal data. However, I have concerns about construct conceptualization and the introduction that limit my enthusiasm about the manuscript in its current form. Below I offer some suggestions that I hope will help this work move forward.

Authors' response: Thank you for taking the time to review our manuscript. We appreciate your encouragement and the constructive suggestions you've provided to further enhance the quality of our paper.

2. **Reviewer's comment:** "I have concerns with authors' characterization of genuine emotional expression as a form of emotion regulation. Authors introduce genuine emotional expression as a specific form of emotion regulation in parenting on p. 1, but do not provide citations for this claim. In reviews of parental emotion regulation (Zhang et al., 2023 <https://doi.org/10.1016/j.dr.2023.101092>; Zimmer-Gembeck et al., 2021 <https://doi.org/10.1177/01650254211051086>) and scales assessing parental emotion regulation (Pereira et al., 2017 <https://doi.org/10.1007/s10826-017-0847-9>; Rodriguez & Shaffer, 2021 <http://doi.org/10.1037/fam0000808>), genuine emotional expression is not mentioned. Further, in models of parental emotion socialization by Nancy Eisenberg and Amanda Morris (see also Hajal & Paley, 2020 <https://doi.org/10.1037/dev0000864>; Edler & Valentino, 2024 <https://doi.org/10.1037/bul0000423>), parental emotional expressiveness is conceptualized as a behavior distinct from parental emotion regulation. In a revision, authors could remove mentions of genuine emotional expression as a form of emotion regulation. Indeed, the cited Yang et al. (2019) study was focused on emotional labor in the workplace; I do not believe that Yang and colleagues suggested that genuine emotional expression was a form of emotion regulation."

"Authors may have a different perspective and are welcome to present it in their manuscript. That said, the proposal that genuine emotional expression is a form of emotion regulation should be supported by a more extensive argument and citations. Or, the claim could be stated explicitly as a novel proposal. Another idea is for authors to conceptualize their construct as a lack of expressive suppression. This would cohere with the literature on parental emotion regulation, James Gross' key process model of emotion regulation, and the meta-analysis by Brandão et al. (2024) that authors cite."

"Regardless of whether authors continue to (1) refer to their construct as genuine emotional expression and (2) claim it is a form of emotion regulation, they also must contend with the limitation that the items do not capture the valence or extremity of emotional expressions (in contrast, Halberstadt's oft-used 1995 Self-Expressiveness in the Family Questionnaire <https://doi.org/10.1037/1040-3590.7.1.93> distinguishes between positive and negative emotional expressiveness). On p. 5 authors state that "genuine expression can be considered an adaptive emotion regulation strategy if habitually used." What if parents are expressing negative emotions? There are a number of studies connecting parents' frequent/high levels of negative emotional expressiveness to poor outcomes for youth (see meta-analysis by Zinsser et al 2021 <https://doi.org/10.1016/j.ecresq.2021.02.001>)."

Authors' response: Thank you for your thoughtful and constructive feedback. We have revised the conceptualization of genuine expression in both the Introduction and Theoretical Background sections. Additionally, we have addressed concerns regarding the measurement of this construct in the Limitations section.

We draw on both the emotional labor and emotion socialization frameworks. In the emotional labor literature, genuine expression refers to the authentic display of one's true emotions in a socially appropriate manner, regardless of emotional valence. This framework, originally developed in occupational contexts, has been adapted by several scholars to the parenting domain. The measure we used for genuine expression is rooted in this tradition. We fully acknowledge, however, that the measure may not be optimal and that there is a discrepancy between the conceptual definition and its operationalization.

From the emotion socialization perspective, parental emotional expression can be considered an emotion-related parenting strategy depending on the underlying goal. This view is also applicable to our study. Therefore, we have integrated both perspectives to provide a more comprehensive conceptualization. We are grateful for the references you suggested, which have been incorporated into our revised manuscript.

We now cautiously propose that habitual genuine expression may serve as an adaptive emotion-related parenting strategy for parents, as it supports authenticity and conserves emotional resources. However, its adaptiveness for children likely depends on the parent's ability to express emotions in a skillful and sensitive manner. When parents communicate their genuine emotions constructively and in developmentally appropriate ways, rather than expressing anger or sadness without reflection, it can foster positive outcomes for children. In this way, genuine expression, when enacted with care and competence, can serve both parent- and child-focused functions.

Changes in the manuscript:

Pages 3-4

"This study aimed to address these gaps by focusing on the dynamic interplay between parental burnout and genuine emotional expression, an often-overlooked construct in parenting literature. Parental genuine expression is defined as the open yet appropriate communication of one's true feelings to their children, regardless of affective valence (Humphrey et al., 2015; Yang et al., 2019). Drawing on emotional labor (Glomb & Tews, 2004; Hochschild, 1983; Humphrey et al., 2015) and emotion socialization (Morris et al., 2007) frameworks, we suggest that, when enacted deliberately and skillfully, genuine expression serves as an adaptive form of emotion regulation in parent-child interactions, fulfilling both parent- and child-focused functions."

Pages 5-7

"Among others, managing the expression of emotions is an integral part of emotion regulation (Gross, 1998b). When such regulation is directed toward achieving interpersonal goals, it is also known as emotional labor (Hochschild, 1983). The present study sheds light on genuine expression, a dimension of emotional labor, defined as the appropriate and authentic display of one's true emotions toward others (Glomb & Tews, 2004; Yang et al., 2019). Initially conceptualized in occupational settings, emotional labor encompassed two strategies: surface acting (i.e., suppressing undesired emotions or faking desired emotions) and deep acting (i.e., attempting to change feelings to produce a more genuine display). Subsequent research work expanded this framework to include genuine expression as a third strategy (Glomb & Tews, 2004; Yang et al., 2019). Scholars have argued that emotional labor constitutes a form of emotion regulation aimed at interpersonal goals (Grandey, 2000; Grandey & Melloy, 2017; Schrodt & O'Mara, 2019) and that this concept can be meaningfully applied to parenting (Lin et al., 2021; Schrodt & O'Mara, 2019). According to Ashforth and Humphrey (1993), the heart of emotional labor lies in expressing emotions in accordance with organizational or social norms; thus, even when parents genuinely feel enthusiastic or sad and express these emotions appropriately to their children, they are still engaging in deliberate emotional management. Parental emotional expression is also central to the emotional socialization of children (Morris et al., 2007). In this literature, parents' emotional expressions serve specific emotion-related or childrearing goals, such as relieving parents' own physiological arousal, modeling the acceptability of emotional experience and expression, or guiding children's behavior in particular situations. Morris et al. (2007) further suggest that mild and moderate degrees of expression of negative expressions can aid children in learning about emotions and emotion regulation. Consequently, how parents express emotions to their children is not only crucial for adapting to dynamic parent-child interactions but also plays a fundamental role in shaping children's emotional development and

long-term adjustment (Morris et al., 2007). Not least, genuine expression is considered resource-conserving, as it fosters congruence between parents' inner-self and outer-behaviors (English & John, 2013) and allows parents to communicate their authentic selves to their children, conveying who they are, what they value and desire, and how they are connected to their children (Humphrey et al., 2015). This perspective aligns with the organismic view of wellness on emotion regulation, which holds that emotion expression enhances well-being and mental health, and strengthens resilience to stress when it supports personal authenticity and autonomous choice (Roth et al., 2019). Thus, genuine expression (if habitually used) can be considered an adaptive emotion-related parenting strategy for parents, as it conserves resources and supports authenticity. However, its adaptiveness for children may depend on the parent's ability to express emotions skillfully and sensitively. When parents communicate their genuine emotions in a constructive and developmentally appropriate manner, rather than expressing anger or sadness without reflection or guidance, it can foster positive outcomes for children. In this way, genuine expression, when enacted with skill, serves both parent- and child-focused functions."

Pages 29-30

"First and foremost, although genuine expression can theoretically be considered a parenting strategy related to emotion regulation, the measure used in this study did not capture several important dimensions: the valence of the expressed emotion (i.e., whether it was positive or negative), the context sensitivity of expression (e.g., whether emotions were expressed in a reflective and appropriate manner), or the explicit goals underlying these expressions (e.g., signaling that experiencing emotions is acceptable). This is a notable limitation, particularly given research showing that frequent expression of negative emotions by parents can be associated with adverse outcomes for children (Zinsser et al., 2021). To advance understanding of genuine expression as a regulatory strategy in parenting, future research should differentiate between the types of emotions expressed, the sensitivity of their expression, and the underlying goals."

3. **Reviewer's comment: Abstract**

Authors write "Dynamic structural equation modeling (DSEM) was used to test reciprocal within-person relations between both constructs over time, to assess individual differences in dynamic patterns, and to explore whether these patterns played a mediating role in changes from baseline to follow-up." Authors should state the constructs in which changes from baseline to follow-up are considered.

Authors' response: Thank you for your comment. We have specified the changes accordingly.

Changes made in the manuscript: Page 2

"Dynamic structural equation modeling (DSEM) was used to test reciprocal within-person relations between both constructs over time, to assess individual differences in dynamic patterns, and to explore whether these patterns mediated changes in burnout and expression from baseline to follow-up."

4. **Reviewer's comment: Abstract**

Authors state that findings "highlight the importance of personalized, real-time interventions that target emotional regulation and burnout recovery in parents ..." This could be taken to imply that the present study analyzed an intervention; rewording can clarify that the study has implications for the design of intervention programs.

Authors' response: Thanks a lot for your suggestion. We have revised accordingly.

Changes made in the manuscript: Page 2

"They suggest that future intervention programs could benefit from being personalized and delivered in real time, targeting emotional regulation and burnout recovery in parents, particularly during emotionally intense periods such as the holiday season."

5. **Reviewer's comment: Introduction**

On p. 3, authors use Brandão et al. (2024) as a citation for "the profound influences of emotion regulation on mental health". The opening to the manuscript would be even stronger if authors

instead reference Brandão for their meta-analysis of the association between parental burnout and parental emotion regulation (Brandão's analyses do not examine parental burnout and parental mental health associations). Similarly, authors write on p. 4 that parental burnout is linked to "increased likelihood of parental neglect or violence towards children (Brianda et al., 2020; Kawamoto et al., 2018)"; Kawamoto et al. (2018) did not examine parental neglect or violence as an outcome.

Authors' response: Thank you for raising this important point. We have now cited two meta-analyses on the effects of emotion regulation on mental health (Aldao et al., 2010; Webb et al., 2012) and corrected the references related to parental neglect and violence (Mikolajczak, Brianda, et al., 2018; Mikolajczak, Raes, et al., 2018).

Changes in the manuscript:

Page 3

"At the intersection of parenting and affective science, the link between parental burnout and emotion regulation has garnered growing scholarly attention, driven by the alarming prevalence of parental burnout (Roskam et al., 2021, 2022) and the profound influences of emotion regulation on mental health (e.g., Aldao et al., 2010; Brandão et al., 2024; Webb et al., 2012)."

Page 5

"It is linked to more frequent and intense partner conflicts, which heighten marital distress (Mikolajczak, Raes, et al., 2018), as well as increased likelihood of parental neglect or violence towards children (Mikolajczak, Brianda, et al., 2018; Mikolajczak, Raes, et al., 2018)."

6. **Reviewer's comment: Introduction**

On p. 3, authors write: "However, the directionality of these variables remains empirically inconsistent..."; authors likely mean the directionality of effect between these variables remains empirically inconsistent.

Authors' response: Yes, you are right. We have revised it accordingly.

Changes in the manuscript: Page 3

"However, the directionality of effects between these variables remains empirically inconsistent, particularly when differentiating between within- and between-person levels (Blanchard et al., 2025; Brandão et al., 2024; Lin et al., 2023; Teuber et al., 2025)."

7. **Reviewer's comment: Introduction**

On p. 4, authors use causal language in the following sentences: "If left untreated, parental burnout can lead to more severe mental issues, including depression, suicidal ideation, and substance abuse (Brianda et al., 2020; Kawamoto et al., 2018). It also disrupts biological processes and contributes to somatic complaints and sleep issues (Mikolajczak et al., 2019). Beyond these negative influences on health, parental burnout impairs family functioning and child development." In the case that any of these studies were nonexperimental, causality should not be implied.

Authors' response: Thank you for pointing this out. We agree that the original phrasing could imply causality, whereas the cited studies are primarily correlational. We have revised the text to emphasize associations rather than causal effects. Specifically, we now state that parental burnout is "associated with" more severe mental health issues, disruptions in biological processes, somatic complaints, sleep problems, and impairments in family functioning and child development.

Changes in the manuscript: Pages 4-5

"Parental burnout has been associated with more severe mental issues, including depression, suicidal ideation, and substance abuse (Mikolajczak et al., 2023; Paula et al., 2022; Ren et al., 2024). It is also linked to disruptions in biological processes and to somatic complaints and sleep issues (Brianda et al., 2020; Mikolajczak et al., 2019). Beyond these negative associations with health, parental burnout is further related to impairments in family functioning and child

development. It is linked to more frequent and intense partner conflicts, which heighten marital distress (Mikolajczak, Raes, et al., 2018), as well as increased likelihood of parental neglect or violence towards children (Mikolajczak, Brianda, et al., 2018; Mikolajczak, Raes, et al., 2018)."

8. **Reviewer's comment: Introduction**

A notable strength of this study is its ability to disaggregate between within- and between-person effects. On p. 6, authors do a nice job describing that prior studies on parental burnout and parental emotion regulation have generally not been longitudinal or used methods that disaggregate these between/within effects. For readers less familiar with the distinction between within- and between-person effects, authors could provide an example of each.

Authors' response: Thank you for this helpful suggestion. We have added concise examples to illustrate the distinction between within-person and between-person effects for readers who may be less familiar with these concepts.

Changes in the manuscript: Page 7

"Moreover, this assumption was derived from between-person analyses, which compare different individuals. For example, between-person effects might show that parents who generally report higher use of reappraisal also tend to report lower burnout than other parents. In contrast, within-person effects track changes over time within the same individual, such as whether a parent experiences lower burnout on days when they engage more in reappraisal than they usually do themselves."

9. **Reviewer's comment: Introduction**

On p. 7 authors write that "Affective dynamics are generally considered adaptive"—this is a broad statement that authors contradict when they go on to write "affective inertia is commonly linked to psychological maladjustment"; rewording will fix this.

Authors' response: Thank you for your suggestion. We agree that the original phrasing was contradictory and have reworded it to ensure clarity and consistency.

Changes in the manuscript: Page 9

"Affective dynamics are considered adaptive when they enable individuals to respond flexibly to changing environmental demands and internal regulatory processes (Houben et al., 2015; Koval et al., 2016)."

10. **Reviewer's comment: Introduction**

On p. 8 authors describe the interesting findings from Blanchard et al. (2025) and say that "They found that inertia was the most robust predictor." Can authors describe this finding in more detail? Was inertia in parental burnout positively associated with overall burnout severity (or negatively)?

Authors' response: Thank you for your suggestion. We have expanded this section to provide more detail on the findings of Blanchard et al. (2025), including the direction of the association between inertia in parental burnout and overall burnout severity.

Changes in the manuscript: Page 10

"They found that inertia of emotional distancing was the most robust positive predictor and interpreted this to mean that parents suffering from more severe burnout are more likely to remain trapped in a state of emotional distance from their children."

11. **Reviewer's comment: Introduction**

On p. 8-9 authors state that genuine expression "initially reflects the positive side of emotional labor (Humphrey et al., 2015)." It is not clear what authors mean, perhaps they should define emotional labor. The following sentence "While inertia is more commonly studied in negatively valenced affects, more positively valenced affective processes like genuine expression are not as well understood, especially whether these positively valenced processes are (mal)adaptive for psychological adjustment." also could be clearer.

Authors' response: Thank you for this helpful suggestion. We agree that the initial description of emotional labor lacked clarity. In the revised manuscript, we now briefly define emotional labor and mention its three dimensions: surface acting, deep acting, and genuine expression. We also clarify that, within this framework, genuine expression is considered the most functional dimension compared to the other two.

Changes in the manuscript: Page 10

“Compared to parental burnout, much less known is about the dynamic patterns of genuine expression, which initially reflects the most functional dimension of emotional labor (Humphrey et al., 2015).”

12. **Reviewer's comment: Introduction**

When authors write on p. 9 that “Findings on whether variability benefits psychological adjustment are mixed, ranging from negative links (Blanke et al., 2020; Elkjær et al., 2022), to non-significant findings (Wang et al., 2021), and even positive associations (Aldao & Nolen-Hoeksema, 2013)”, it seems relevant what the construct is. Variability in parental burnout? Variability in genuine emotional expression? Variability in and of itself is not meaningful without knowing the construct. I tried to look up these papers but none are in the reference section. I noticed Lin et al. (2023) and English & John (2013) are also missing from the reference section.

Authors' response: Thank you for these important comments. The variability we referred to concerns the use of specific emotion regulation strategies, and we have clarified this in the revised text. We also identified and corrected issues with the reference list, which were caused by a problem when updating the bibliography using the Zotero add-on in Google Docs. We have now carefully reviewed and updated all references, including the missing citations you noted.

Changes in the manuscript: Page 10

“Findings on whether variability in the use of a specific emotion regulation strategy benefits psychological adjustment are mixed, ranging from negative links (Blanke et al., 2020; Elkjær et al., 2022), to non-significant findings (Wang et al., 2021), and even positive associations (Aldao & Nolen-Hoeksema, 2013).”

13. **Reviewer's comment: Current Study/Method/Data Analysis/Results**

These sections were well-written. Authors do a good job explaining the dynamics of interest (inertia, variability, person-specific means) and DSEM. Their writeup of the results was overall clear. I only have a few suggestions.

Authors should describe what a random effect is in the context of a DSEM for less familiar readers.

Authors' response: Thank you for your positive feedback and for acknowledging the clarity of our explanation of affective dynamics and DSEM. We appreciate your recognition of the effort invested in presenting these sections clearly.

We have added a clear description of random effects in the context of DSEM for readers who may be less familiar with the approach. Specifically, we now explain that in DSEM, random effects refer to parameters (e.g., autoregressive or cross-lagged effects) that are allowed to vary across individuals. This approach captures between-person differences in the strength or direction of these effects, rather than assuming a uniform effect across all participants.

Changes in the manuscript: Page 17, Footnote 1

“Within DSEM, random effects refer to parameters that are allowed to vary across individuals, capturing between-person differences in the strength or direction of the effects.”

14. **Reviewer's comment: Current Study/Method/Data Analysis/Results**

As DSEM is still a relatively new method, I wonder if authors could include their equations and a figure representation of the model (as in Hamaker et al., 2018 <https://doi.org/10.1080/00273171.2018.1446819> and McNeish & Hamaker, 2020 <http://doi.org/10.1037/met0000250>).

15. **Reviewer's comment: Current Study/Method/Data Analysis/Results**

On p. 19, authors state that “parental gender showed a non-null effect ($B = .152$ [.007, .299])” on follow-up parental burnout. Back on p. 13 authors state how gender was dummy-coded, but it should be restated in the results or reworded to indicate that mothers vs. fathers were reporting more burnout at the follow-up assessment.

Authors' response: Thank you for your comment. We have revised the text to restate the coding and clarify the interpretation.

Changes in the manuscript: Page 21

“Among these covariates, only parental gender showed a non-null effect ($B = .152$ [.007, .299]), suggesting that at follow-up, mothers reported higher levels of burnout compared with fathers.”

16. **Reviewer's comment: Discussion**

Clear interpretation of the within-person cross-lagged effect. Good explanation of how both inertia and variability can be elevated. A key limitation to note in the limitations/future directions section is that the items for genuine emotional expression do not capture the valence or extremity of emotional expressions.

Authors' response: Thank you for your comment. We have added this important point as a limitation in the Limitations section, noting that the items for genuine emotional expression do not capture the valence or extremity of expressed emotions, the context sensitivity of expression, or the explicit goals underlying these expressions.

Changes in the manuscript: Pages 29-30

“First and foremost, although genuine expression can theoretically be considered a parenting strategy related to emotion regulation, the measure used in this study did not capture several important dimensions: the valence of the expressed emotion (i.e., whether it was positive or negative), the context sensitivity of expression (e.g., whether emotions were expressed in a reflective and appropriate manner), or the explicit goals underlying these expressions (e.g., signaling that experiencing emotions is acceptable). This is a notable limitation, particularly given research showing that frequent expression of negative emotions by parents can be associated with adverse outcomes for children (Zinsser et al., 2021). To advance understanding of genuine expression as a regulatory strategy in parenting, future research should differentiate between the types of emotions expressed, the sensitivity of their expression, and the underlying goals.”

Reviewer #2

1. **Reviewer's comment: SUMMARY**

The Authors investigate the relationship between parental burnout and genuine expression during the Christmas period in a large sample of parents in the UK, using a pre-existing ESM dataset. The authors focused on two key research questions: 1. Is there a reciprocal relationship between parental burnout and genuine expression (within-person). 2. Do individual differences in within person patterns (inertia, variability and person mean) of parental burnout and genuine expression mediate their association from baseline to follow-up.

I found the research question worthy of exploring (although I am confused by some of the literature used to justify the investigation), and the introduction is generally well written, expressing clear arguments. The method matches the research question and is generally clearly explained. A strong point of the paper is the clarity with which the authors describe DSEM— very well done there. Yet, I have some comments concerning OSF practices, results and interpretations. I hope the authors will take the comments as opportunities to strengthen their work, and I want to highlight that I genuinely enjoyed reading their paper.

Authors' response: Thank you for taking the time to review our manuscript. We appreciate your encouragement and the constructive suggestions you've provided to further enhance the quality of our paper.

2. **Reviewer's comment:** I am confused by the framing of genuine expression as an “emotion regulation strategy” (line 92) in the context of this specific paper. While the points made on page 5 are understood and potentially valid, to my understanding of the Process Model of ER (Gross, 1998; 2015), Affect/Emotion Regulation strategies are implemented to increase, decrease, or maintain emotions. This aspect seems to be mostly missing from the conceptualization offered in the introduction, and no reference is made to the IER literature—this literature would have been perhaps more directly relevant, considering that the Authors seems to allude to the fact that in the parenting context, emotion expression is fundamental in shaping children’s emotional socialization and long-term development (line 82-84). My confusion is also matched by the methodological operationalization of the “emotion regulation strategy”. When looking at the phrasing of the item (line 287-294), there is no indication of this variable being “a strategy” (i.e., used for the intent of regulating one’s emotions, or their children’s emotion).

Authors' response: Thank you for your comment, which aligns with one of Reviewer 1’s major concerns. We have substantially revised the conceptualization of genuine expression in both the Introduction and Theoretical Background sections and addressed concerns regarding its measurement in the Limitations section.

We draw on both the emotional labor and emotion socialization frameworks. In the emotional labor literature, genuine expression refers to the authentic display of one’s true emotions in a socially appropriate manner, regardless of emotional valence. This framework, originally developed in occupational contexts, has been adapted by several scholars to the parenting domain. The measure we used for genuine expression is rooted in this tradition. We fully acknowledge, however, that the measure may not be optimal and that there is a discrepancy between the conceptual definition and its operationalization.

From the emotion socialization perspective, parental emotional expression can be considered an emotion-related parenting strategy depending on the underlying goal. This view is also applicable to our study. Therefore, we have integrated both perspectives to provide a more comprehensive conceptualization. We are grateful for the references you suggested, which have been incorporated into our revised manuscript.

We now cautiously propose that habitual genuine expression may serve as an adaptive emotion-related parenting strategy for parents, as it supports authenticity and conserves emotional resources. However, its adaptiveness for children likely depends on the parent’s ability to express emotions in a skillful and sensitive manner. When parents communicate their genuine emotions constructively and in developmentally appropriate ways, rather than expressing anger or sadness without reflection, it can foster positive outcomes for children. In this way, genuine expression, when enacted with care and competence, can serve both parent- and child-focused functions.

**Changes in the manuscript:
Pages 3-4**

“This study aimed to address these gaps by focusing on the dynamic interplay between parental burnout and genuine emotional expression, an often-overlooked construct in the parenting literature. Parental genuine expression is defined as the open yet appropriate communication of one’s true feelings to their children, regardless of affective valence (Humphrey et al., 2015; Yang et al., 2019). Drawing on emotional labor (Glomb & Tews, 2004; Hochschild, 1983; Humphrey et al., 2015) and emotion socialization (Morris et al., 2007) frameworks, we suggest that, when enacted deliberately and skillfully, genuine expression serves as an adaptive form of emotion regulation in parent-child interactions, fulfilling both parent- and child-focused functions.”

Pages 5-7

“Among others, managing the expression of emotions is an integral part of emotion regulation (Gross, 1998b). When such regulation is directed toward achieving interpersonal goals, it is also known as emotional labor (Hochschild, 1983). The present study sheds light on genuine expression, a dimension of emotional labor, defined as the appropriate and authentic display of one’s true emotions toward others (Glomb & Tews, 2004; Yang et al., 2019). Initially conceptualized in occupational settings, emotional labor encompassed two strategies: surface

acting (i.e., suppressing undesired emotions or faking desired emotions) and deep acting (i.e., attempting to change feelings to produce a more genuine display). Subsequent research work expanded this framework to include genuine expression as a third strategy (Glomb & Tews, 2004; Yang et al., 2019). Scholars have argued that emotional labor constitutes a form of emotion regulation aimed at interpersonal goals (Grandey, 2000; Grandey & Melloy, 2017; Schrodt & O'Mara, 2019) and that this concept can be meaningfully applied to parenting (Lin et al., 2021; Schrodt & O'Mara, 2019). According to Ashforth and Humphrey (1993), the heart of emotional labor lies in expressing emotions in accordance with organizational or social norms; thus, even when parents genuinely feel enthusiastic or sad and express these emotions appropriately to their children, they are still engaging in deliberate emotional management. Parental emotional expression is also central to the emotion socialization of children (Morris et al., 2007). In this literature, parents' emotional expressions serve specific emotion-related or childrearing goals, such as relieving parents' own physiological arousal, modeling the acceptability of emotional experience and expression, or guiding children's behavior in particular situations. Morris et al. (2007) further suggest that mild and moderate degrees of expression of negative expressions can aid children in learning about emotions and emotion regulation. Consequently, how parents express emotions to their children is not only crucial for adapting to dynamic parent-child interactions but also plays a fundamental role in shaping children's emotional development and long-term adjustment (Morris et al., 2007). Not least, genuine expression is considered resource-conserving, as it fosters congruence between parents' inner-self and outer-behaviors (English & John, 2013) and allows parents to communicate their authentic selves to their children, conveying who they are, what they value and desire, and how they are connected to their children (Humphrey et al., 2015). This perspective aligns with the organismic view of wellness on emotion regulation, which holds that emotion expression enhances well-being and mental health, and strengthens resilience to stress when it supports personal authenticity and autonomous choice (Roth et al., 2019). Thus, genuine expression (if habitually used) can be considered an adaptive emotion-related parenting strategy for parents, as it conserves resources and supports authenticity. However, its adaptiveness for children may depend on the parent's ability to express emotions skillfully and sensitively. When parents communicate their genuine emotions in a constructive and developmentally appropriate manner, rather than expressing anger or sadness without reflection or guidance, it can foster positive outcomes for children. In this way, genuine expression, when enacted with skill, serves both parent- and child-focused functions."

Pages 29-30

"First and foremost, although genuine expression can theoretically be considered a parenting strategy related to emotion regulation, the measure used in this study did not capture several important dimensions: the valence of the expressed emotion (i.e., whether it was positive or negative), the context sensitivity of expression (e.g., whether emotions were expressed in a reflective and appropriate manner), or the explicit goals underlying these expressions (e.g., signaling that experiencing emotions is acceptable). This is a notable limitation, particularly given research showing that frequent expression of negative emotions by parents can be associated with adverse outcomes for children (Zinsser et al., 2021). To advance understanding of genuine expression as a regulatory strategy in parenting, future research should differentiate between the types of emotions expressed, the sensitivity of their expression, and the underlying goals."

3. **Reviewer's comment:** Could authors please report correlations (within and between person) for the variables of interest (in the manuscript, or in the supplementary materials). Did the authors check for multicollinearity?

Authors' response: Thank you for your comment. We have now reported the correlations for the variables of interest (Table 2). Regarding multicollinearity, we examined the correlations and found them to be moderate overall. Given this and the conceptual distinctiveness of the variables, we did not anticipate multicollinearity issues. Furthermore, we calculated the VIF values for the between-level predictors, and they were all quite small and well below the typical cutoff of >5 where multicollinearity is a concern (range = 1.10 to 1.88).

Variable	R ²	VIF	Model
Mother	.089	1.10	Probit
Needs	.113	1.13	Probit
Single	.469	1.88	Probit
Income	.096	1.11	Continuous
PBT1	.169	1.20	Continuous
EXPT1	.183	1.22	Continuous

Changes in the manuscript:

Page 18

“At the within-person level, after latent decomposition, burnout and expression were moderately correlated ($r = -.33$, $SE = .03$, $p < .001$). Correlations among the variables of interest at the between-person level following latent decomposition are presented in Table 2.”

Table 1 and Table 2

Table 1

Mean (M) and Standard Deviations (SD) of Parental Burnout and Genuine Expression Across the Experience Sampling Period, Baseline, and Follow-Up after Latent Decomposition using Mplus (TYPE = TWOLEVEL)

	Experience Sampling Period		Baseline M (SD)	Follow-up M (SD)
	Between-Person M (SD)	Within-Person M (SD)		
Parental Burnout	2.06 (1.00)	0 (.72)	2.70 (1.28)	2.55 (1.20)
Genuine Expression	5.78 (.93)	0 (.99)	5.76 (1.12)	5.64 (1.10)

Note. Latent decomposition was performed only for the experience sampling measures.

Table 2

Correlations of Variables of Interest at the Between-Person Level after Latent Decomposition using Mplus (TYPE = TWOLEVEL)

Variable	1	2	3	4	5	6	7	8	9
1 Mother									
2 Needs	.06								
3 Single parenthood	.12*	.21**							
4 Income	-.07	-.14*	-.28***						
5 Burnout baseline	.07	.05	.09	-.05					
6 Expression baseline	.09	-.10	-.04	.10	-.39***				
7 Burnout follow-up	.10	.04	.04	-.08	.75***	-.36***			
8 Expression follow-up	.10	-.02	.06	.10	-.36***	.47***	-.46***		
9 Burnout ESM	-.02	.11	.01	-.12	.59***	-.28***	.67***	-.39***	
10 Expression ESM	.08	-.07	.06	.05	-.40***	.43***	-.50***	.54***	-.50***

Note. * $p < .05$, ** $p < .01$, *** $p < .001$. Needs = the presence of special needs in children. ESM = measured during the experience sampling period. Latent decomposition was performed only for the experience sampling measures.

4. **Reviewer’s comment:** The data collection of this project was pre-registered, however there is no reference in the pre-registration speaking for the “conceptual alignment of the hypotheses with the overarching aims of the broader project” (line 210-212). The variables and hypotheses were not explicitly stated in the pre-registration. The code available on the OSF is difficult to navigate, even more so without the data available to reproduce the analysis. Perhaps authors could knit a document and upload to the OSF if data cannot be made available on initial submission? No information on the data preprocessing and the decision-making concerning rationale for exclusions is provided. This is a problem stemming from the quality of the data collection pre-registration, and flowing on to the level of information provided in the manuscript (and

supplementary materials). There was no rationale for data collection sample size (e.g. power). Unfortunately, I cannot consider this paper to be aligned with best open science practices.

Authors' response: Thank you for this detailed and constructive feedback. We would like to clarify the following points:

Preregistration and Hypotheses: Our preregistration primarily focused on the associations between emotional labor and parental burnout (e.g., H2c: The amount of emotional labor positively predicts parental burnout). We also indicated in the preregistration that the study included exploratory components, as noted in the section on Exploratory Analysis. While we mentioned reciprocal relationships, we acknowledge that the preregistration did not provide detailed information on the subdimensions of emotional labor (e.g., genuine expression) or on the dynamic parameters. These hypotheses were developed later, informed by a deeper engagement with the literature and inspired by recent work (e.g., Blanchard et al., 2025). We recognize that preregistering every possible hypothesis in a complex, multi-construct project is challenging, and we believe that the field would benefit from collective efforts to establish transparent yet practical preregistration standards.

Data and Code Transparency: We appreciate your suggestion regarding code accessibility. We have now commented on the Mplus input file more structurally and detailly. We have also made our data publicly accessible.

Sample Size and Power Analysis: Regarding sample size, as stated in the preregistration, we did not conduct an a priori power analysis because our recruitment strategy aimed to include as many parents as possible within the available resources. We intentionally did not include a power analysis in the manuscript to avoid creating the impression that it was conducted prior to data collection.

However, our sample size ($N = 293$) has adequate statistical power; this conclusion can be supported by the following two arguments:

- (1) Sultzberg and Muthén (2018) conducted power simulations for common designs to suggest general guidelines for adequate sample sizes. The focal effects in our study correspond to Model 6 (between-person predictors of a residual variance) or Model 9 (residual variance as a mediator) in their Figure 1. According to Figure 19, for 50 time points per person, adequate power to detect between-person effects on a log-residual variance is achieved with $N = 100$, well below our sample size of 293. Similarly, Figure 20 suggests that detecting between-person mediation involving a log-residual variance requires $N = 250$, which is again lower than our sample size.
- (2) Most of the effects of interest were plausibly non-null (i.e., 0 was not in the credible interval). If the model detected these effects, there are only two possibilities: (a) we had sufficient power, or (b) we made a Type I error. Conducting a post hoc power analysis would not change this interpretation.

We appreciate your emphasis on open science practices and will take these points into account to strengthen the transparency and reproducibility of our work.

Reference: Schultzberg, M., & Muthén, B. (2018). Number of subjects and time points needed for multilevel time-series analysis: A simulation study of dynamic structural equation modeling. *Structural equation modeling: a multidisciplinary journal*, 25(4), 495-515. <https://doi.org/10.1080/10705511.2017.1392862>

Changes in the manuscript: Page 12

“This study was part of a larger, preregistered project investigating parental momentary experiences during the Christmas season. The preregistration protocol is accessible at https://osf.io/4xga6/?view_only=136f8723227148ed84e9f2c8f02eafe6. Although the specific hypotheses for this study were not explicitly preregistered, they were conceptually aligned with the overarching aims of the broader project. As such, the specific hypotheses tested in this study can be considered exploratory. Data and analysis codes associated with this study are available

at https://osf.io/5dnrm/?view_only=dd92d3260d264eb08251c3af32678a92. Ethical approval was obtained from [blinded for peer review]. Participants provided signed informed consent, and participation was voluntary.”

5. Reviewer’s comment: Line 410-420

I do apologize if I somehow missed it, but I am unclear on how exactly the mediation analyses were set up. Then, in the results section, it is not specified if the mediation is partial or full. A figure, and/or a table of results would maybe make it clearer?

Authors’ response: Thank you for pointing this out. We apologize that the description of the mediation analyses was not sufficiently detailed in the original submission. We have now added a clear explanation of how the mediation models were specified in the Data Analysis section. In addition, we have clarified in the Results section whether the observed mediation effects are partial.

Changes in the manuscript:

Page 17

“If these relationships held, we further examined potential mediation effects of these dynamic parameters using the MODEL CONSTRAINT function in Mplus (i.e., axb ; baseline \xrightarrow{a} dynamic parameter \xrightarrow{b} follow-up).”

Page 22

“The presence of a direct link between baseline burnout and burnout person-specific mean across the experience sampling period, as well as between this mean and follow-up burnout, pointed to a potential mediating role of person-specific mean levels. Indeed, our analysis confirmed a partial mediating effect ($B = .288$ [.224, .365]), indicating that participants who started with higher parental burnout at baseline also demonstrated greater average during the experience sampling period, which, in turn, predicted higher burnout at the follow-up after controlling for relevant covariates. A similar partial mediation pattern was found for genuine expression: person-specific mean levels of genuine expression mediated the relationship between baseline and follow-up genuine expression ($B = .168$ [.118, .267]). That is, participants who started with higher levels of genuine expression at baseline also showed higher mean levels during the experience sampling period, which subsequently predicted higher levels of genuine expression at follow-up.”

6. Reviewer’s comment: Line 561-582

Given my other comments about theoretical points, and operationalization of the main variables, I think the recommendations over-reach with interpreting the findings. I recommend the authors remove this part, or significantly tone down the recommendations.

Authors’ response: Thank you for this thoughtful suggestion. We agree with your assessment and have removed the section in question from the revised manuscript.

Changes in the manuscript: Page 28

7. Reviewer’s comment: The paper seems to put a major weight on the fact that data was collected during the festive season. The justification for this point is clear, and is understandable, however, I note that there was no real measure of the impact of the festive season. Accordingly, I think this should be a point that is clearly made in the method, but not leveraged as much in the title, and discussion, to frame the results as if there was a measured role of the festive season, on the relationship between variables.

Authors’ response: Thank you for this comment. We have revised the manuscript to clarify that the festive season was not directly measured and therefore cannot be interpreted as a tested moderator or explanatory factor. Specifically, we have: (1) updated the Method section to note that data collection occurred during the festive season, without implying a measured effect; (2) adjusted the title and to reduce emphasis on the festive season.

However, we decided to retain a brief discussion of the festive season because the timing of data collection provides important context for interpreting the findings and their generalizability. While we did not measure its impact, the festive season is widely recognized as a period associated with increased family obligations, financial strain, and emotional demands, which could plausibly influence parental experiences. Acknowledging this context enhances transparency and helps readers understand potential situational factors that might have shaped the data. Furthermore, noting this limitation can guide future research to examine whether seasonal factors systematically affect emotional labor and parental burnout.

Changes in the manuscript: Title

“Unwrapping the Dynamics of Parental Burnout and Genuine Expression: A Dynamic Structural Equation Modeling Approach”

Page 2

“This study adopted a within-person lens to unpack parental burnout and genuine emotional expression, focusing on their interplay and dynamic patterns – inertia, variability, and person-specific mean – during the Christmas season, an emotionally charged period that offers a valuable time window to study affective dynamics in parenting.”

Page 12

“First, we aimed to examine the temporal, within-person associations between parental burnout and genuine expression during the Christmas festive season in the United Kingdom. Although this time frame may entail additional emotional and logistical demands for parents, the specific impact of the festive season was not focus of this study.”

8. Reviewer’s comment: Line 95

Gross 1998b, 2025, 2024 are cited, but are missing from the reference list.

Authors’ response: Thank you for pointing out these errors. We identified and corrected issues with the reference list, which were caused by a problem when updating the bibliography using the Zotero add-on in Google Docs. We have now carefully reviewed and updated all references, including the missing citations you noted.

9. Reviewer’s comment: Line 105

perhaps another more recent and useful reference for the point the authors make is: Hopwood et al (2025). I think this might also be useful on line 176-177, and in the discussion, if the authors wanted to keep a point about the merits of DSEM in disentangling within from between person effects, and perhaps indicate future directions along the same lines (but as I stated in one of my other points, toning down the claims currently made in the discussion).

Hopwood, C. J., Aafjes-van Doorn, K., Békés, V., Luo, X., Ringwald, W. R., & Wright, A. G. C. (2025). Is psychological research producing the kind of knowledge clinicians find useful? *American Psychologist*. Advance online publication. <https://doi.org/10.1037/amp0001538>

Authors’ response: Thank you for this valuable suggestion. It’s a fascinating study by Hopwood et al. (2025)! Thanks a lot for bringing our attention to it! We have incorporated the reference (Hopwood et al., 2025) to strengthen our discussion of the merits of DSEM in disentangling within- and between-person effects. We also revised the discussion.

Changes in the manuscript: Page 8

“Methodological debates have pointed out that relationships observed at the between-person level do not necessarily hold at the within-person level (Berry & Willoughby, 2017; Hopwood et al., 2025).”

Page 11

“Identifying these associations is not only theoretically important but also has practical relevance for designing personalized interventions (Hopwood et al., 2025).”

Page 30

“Finally, researchers in clinical psychology have emphasized that distinguishing between- and within-person associations is essential for understanding psychopathology and advancing science–practice integration (Hopwood et al., 2025). Building on this insight, we call for greater research efforts using intensive longitudinal designs and DSEM. Such work can provide a more nuanced and actionable understanding of parental burnout, ultimately informing more effective and personalized intervention strategies.”

10. **Reviewer’s comment:** It would have been useful to have an idea of the amount of sample stemming from the same parental dyad.

Authors’ response: Thank you for this valuable comment. We would like to clarify that we did not target parental dyads in our sampling approach and did not collect any identifying information; hence, it is technically possible that two members of the same parental dyad participated in the study, though the method of data collection suggests this is unlikely. For the purposes of interpretation, we treat the data as pertaining to distinct family units.

Changes in the manuscript: Pages 13-14

“After data cleaning, the final sample consisted of 293 participants, which we treated as representing distinct family units (i.e., no parent dyads), with an average age of 38.16 years (SD = 6.92).”

11. **Reviewer’s comment:** The language used in the results section is sometimes confusing as to what exact analysis have been conducted. E.g. line 350: “more likely to report...” Does this mean that analyses were logistic, binarizing the variable (which to my understanding was on a 1-7 scale)? Similar issue with line 357 “they tended to...”. Line 360 “did not lead to subsequent changes...” this is causal language, which is incorrect even when using lagged analyses or DSEM, as the data is still correlational. Line 496-497 “more likely..... less likely...” again, logistic analyses?

Authors’ response: Thank you for pointing out the ambiguity in our wording.

To clarify, the 7-point scale was treated as a continuous variable in our models. This approach is common in psychological research, and recent simulation work suggests that treating 7-point ordinal data as continuous produces estimates that are essentially equivalent to those obtained when modeling the data as categorical, particularly with sample sizes similar to ours (see McNeish & Savord, 2025).

We have further revised the manuscript to make this explicit in the Methods section and clarified the phrasing in the Results section to avoid confusion. We have carefully revised the Results section to ensure that the language accurately reflects the analyses conducted. We also removed any phrasing that could imply causality, such as “did not lead to,” and rephrased these statements to reflect the correlational nature of the data, even when using lagged or DSEM analyses. These changes aim to improve clarity and avoid any misleading interpretations.

Reference: McNeish, D. & Savord, A. (2025). Exploring how many categories are needed to model ordinal intensive longitudinal data as continuous with dynamic structural equation models. *Psychological Methods*. Advance online publication. <https://doi.org/10.1037/met0000784>

Changes in the manuscript: 19

“The results of DSEM showed that both parental burnout ($B = .338$ [.285, .389]) and genuine expression ($B = .205$ [.148, .259]) exhibited non-null autoregressive effects over time at the within-person level. This indicated that individuals were likely to report higher levels of parental burnout or genuine expression at one moment if they had experienced elevated levels of the same construct at an earlier moment, relative to their own average. In Hypothesis 1, we expected negative reciprocal relationships between fluctuations in parental burnout and genuine expression at the within-person level. The results partially supported this hypothesis: fluctuations

in genuine expression were negatively predicted by prior fluctuations in parental burnout: $B = -.099$ $[-.132, -.066]$. In other words, when participants experienced higher parental burnout relative to their own average at a given moment, they are likely to report lower genuine expression relative to their own average at the subsequent moment. Contrary to our expectations, however, the reverse cross-lagged effect was likely negligible, with a posterior distribution centered near zero ($B = -.004$ $[-.015, .006]$) and substantial probability mass around the null. That is, experiencing higher or lower levels of genuine expression than usual at one moment did not predict subsequent changes in parental burnout relative to one's own average."

12. Reviewer's comment: Line 504

"... inertia is less pronounced for understanding the dynamics of positively valenced emotional processes like genuine expression": this statement makes an assumption that people would be only expressing positive emotions genuinely.

Authors' response: Thank you for pointing this out. We agree that the original phrasing could imply that genuine expression pertains only to positive emotions, which was not our intention. We have revised the statement to clarify that genuine expression can involve any type of emotion, positive or negative.

Changes in the manuscript: Page 26

"While affective inertia has been linked to maladaptive processes such as rumination, emotional rigidity, and psychological maladjustment (Koval et al., 2016; Kuppens et al., 2010), the absence of a link here may indicate that inertia is less pronounced for understanding the dynamics of emotional processes like genuine expression (of any emotions). Unlike rigid affective patterns, genuine expression involves open yet context-appropriate communication of emotions, which likely requires adaptability and flexibility on the part of parents."

13. Reviewer's comment: Line 556-561: please provide citations and references for the statements made.

Authors' response: Thank you! We have incorporated references that support our assumption.

Changes in the manuscript: Page 28

"Despite its potential short-term utility, the prolonged suppression of genuine emotions can lead to significant long-term consequences (Chervonsky & Hunt, 2017; Ruan et al., 2020)."